# Human DNA polymerase ε is a source of C>T mutations at CpG dinucleotides

**Marketa Tomkova** [1,4] ✉, **Michael John McClellan** [1,4] ✉, **Gilles Crevel**[2], **Akbar Muhammed Shahid**[1], **Nandini Mozumdar**[1], **Jakub Tomek** [3], **Emelie Shepherd**[1], **Sue Cotterill**[2], **Benjamin Schuster-Böckler** [1,5] ✉ & **Skirmantas Kriaucionis** [1,5] ✉

C-to-T transitions in CpG dinucleotides are the most prevalent mutations in human cancers and genetic diseases. These mutations have been attributed to deamination of 5-methylcytosine (5mC), an epigenetic modification found on CpGs. We recently linked CpG>TpG mutations to replication and hypothesized that errors introduced by polymerase ε (Pol ε) may represent an alternative source of mutations. Here we present a new method called polymerase error rate sequencing (PER-seq) to measure the error spectrum of DNA polymerases in isolation. We find that the most common human cancer-associated Pol ε mutant (P286R) produces an excess of CpG>TpG errors, phenocopying the mutation spectrum of tumors carrying this mutation and deficiencies in mismatch repair. Notably, we also discover that wild-type Pol ε has a sevenfold higher error rate when replicating 5mCpG compared to C in other contexts. Together, our results from PER-seq and human cancers demonstrate that replication errors are a major contributor to CpG>TpG mutagenesis in replicating cells, fundamentally changing our understanding of this important disease-causing mutational mechanism.

The emergence and evolution of tumors are driven by mutations, which can be the result of exogenous or endogenous DNA damage or a product of errors during DNA replication[1,2]. The most common mutation type is a substitution from cytosine to thymine in a CpG dinucleotide (CpG>TpG) across normal somatic and germline cells, as well as cancer cells[3–5]. Germline CpG>TpG mutations are at least ten times more common than expected by chance[6] and represent a frequent cause of many genetic diseases[7,8]. Clustering of cancer mutations into signatures based on the substitution type and context exposed CpG>TpG mutations as the defining feature of somatic single-base substitution signature 1 (SBS1), the most widely observed mutational signature in human cancers and normal cells[4]. Determining the molecular mechanisms that result in CpG>TpG mutations therefore has important implications for our understanding of evolution in populations as well as in cancer.

The elevated CpG>TpG mutation rate has been linked to 5-methylcytosine (5mC), an epigenetic modification that in humans occurs primarily in CpG dinucleotides[9], has an important role in gene regulation and is essential for normal development[10]. It was observed in vitro that 5mC undergoes spontaneous deamination approximately two times faster than unmodified cytosine[11]. Moreover, 5mC deamination produces T, resulting in T:G mismatches, which were shown to be repaired much less efficiently than U:G mismatches created by deamination of unmodified cytosines[12]. CpG>TpG mutations are therefore widely considered to be the result of elevated spontaneous deamination of 5mC.

Surprisingly, we previously observed that CpG>TpG mutations are orders of magnitude more frequent in cancer genomes from individuals with different types of postreplicative mismatch repair

[1]Ludwig Institute for Cancer Research, University of Oxford, Oxford, UK. [2]Molecular and Cellular Sciences, St George's University London, London, UK. [3]Department of Physiology Anatomy and Genetics, University of Oxford, Oxford, UK. [4]These authors contributed equally: Marketa Tomkova, Michael John McClellan. [5]These authors jointly supervised this work: Benjamin Schuster-Böckler, Skirmantas Kriaucionis. ✉e-mail: marketa.tomkova@ludwig.ox.ac.uk; michael.mcclellan@ludwig.ox.ac.uk; benjamin.schuster-boeckler@ludwig.ox.ac.uk; skirmantas.kriaucionis@ludwig.ox.ac.uk

(MMR) deficiency or mutations in the exonuclease domain of the major leading-strand DNA polymerase ε (Pol ε), neither of which were thought to be required for the detection or repair of spontaneous deamination[13]. Instead, MMR and Pol ε 'proofreading' through its exonuclease domain are two key components that repair errors introduced during DNA replication[14], and their defects cause hypermutated tumors in mice[15–20], high mutation burden in yeast[21–23] and the most hypermutated human cancers[24–28]. This led us to hypothesize that CpG>TpG mutations could also be introduced in a deamination-independent manner as a result of polymerase errors during DNA replication.

Error rates of DNA polymerases have previously been measured using mutation-induced loss of activity of reporter genes (hypoxanthine-guanine phosphoribosyltransferase (*HPRT*) and *lacZ*), which can be assayed individually at high scales[29,30]. However, these methods introduce considerable biases as only certain mutations produce a measurable phenotype, leading to poor representation of sequence contexts. Moreover, the effect of cytosine methylation is difficult to study consistently in such cell-based assays.

Here we set out to directly quantify the misincorporation rate and sequence specificity of mutant and wild-type Pol ε using a sequencing-based approach. To exactly determine which template bases result in what misincorporation, we needed a method that can reliably detect mismatched bases in individual molecules of newly synthesized DNA. Standard genome sequencing cannot be used to detect base changes at single-molecule resolution because they cannot distinguish real variants from technical artifacts introduced during library preparation or from base-calling errors by the sequencing pipeline. Several sequencing-based technologies were recently developed to detect very rare variants, including duplex sequencing[31], nanorate sequencing (NanoSeq)[32] or bottleneck sequencing system (BotSeqS)[33]. However, all of them require mutations to be present on both DNA strands, rendering them unsuitable for the direct detection of mismatches introduced by DNA polymerases.

To overcome these limitations, we developed polymerase error rate sequencing (PER-seq), a new method that can detect mismatches introduced by DNA polymerases in a cell-free environment at single-molecule resolution, enabling the quantification of replication errors down to a rate of approximately 1 in $10^6$ replicated bases. We used PER-seq and sequenced over 28 billion bases across more than 130 million molecules to a sufficient depth to detect the misincorporation errors of wild-type and mutant human Pol ε when replicating methylated and unmethylated templates. We show that the sequence-context-specific misincorporation rate of mutant Pol ε measured in vitro closely resembles the mutational signatures observed in tumor samples with combined Pol ε proofreading mutations and MMR deficiency. Strikingly, we detected particularly high Pol ε error rates in a CpG context, which are further increased by the presence of 5mC. Our observations strongly support the hypothesis that CpG>TpG mutations are frequently introduced during DNA replication in a deamination-independent manner.

## Results

### PER-seq

In PER-seq, the template DNA is a 200–300 bp long region of interest (ROI) inserted into a plasmid. Two natural sequences from the human exome were used here (*TP53* and *DNMT1* genes; Methods). Plasmids are first enzymatically purged of DNA damage, one strand of the ROI is then selectively removed and the resulting single-stranded region is filled by a polymerase of interest (based on refs. 34,35; see Methods for more details; Fig. 1a and Extended Data Fig. 1a,b). Mutations in the template and daughter strands of the fully filled plasmids are then determined by an adapted and highly optimized version of maximum depth sequencing[36]. Each ROI-containing molecule receives a unique molecular identifier (DNA barcode), followed by seven to ten rounds of linear amplification, with each linear copy receiving an additional

unique barcode. All DNA fragments are then exponentially amplified, followed by high-throughput Illumina sequencing.

To distinguish true variants (polymerase errors) from false positives (artifacts resulting from DNA amplification, damage or sequencing), we require the variant to be present in at least three independent linear copies. The probability of the same artifact happening independently three times in the same position is $<10^{-9}$ (Supplementary Notes 1 and 2). Indeed, the PER-seq measurements show that the detected mutation frequency is very similar when considering three or more linear copies, whereas only one or two copies are not sufficient to fully distinguish between real variants and false positives (Fig. 1b).

Notably, by also sequencing the template strand of the ROI, we can measure the profile of cytosine deamination and other damage that happened on the template DNA before/during the filling of the gapped plasmid. We can thus subtract these assay-specific artifacts and derive an accurate representation of the type and frequency of mistakes introduced by DNA polymerases (see Methods, Supplementary Notes 2–4, including Fig. 2 under Supplementary Note 2 for assay background estimates). Moreover, this also allows for the subtraction of any potential damage introduced by DNA methyltransferases when methylating DNA templates[37,38].

To validate the method, we introduced predefined single-base variants in the ROI and mixed the mutated plasmids at different dilutions ranging from 1 in 10 to 1 in $10^6$ (Methods). PER-seq discovered the introduced variants at frequencies very close to the expected values (Pearson $R = 0.993$, $P = 4 \times 10^{-199}$; Fig. 1c). Next, we used PER-seq to measure the misincorporation spectrum of *Escherichia coli* DNA polymerase Klenow fragment (Klenow-EXO⁻), a low-fidelity polymerase lacking exonuclease activity. We observed similar frequencies of the individual error types as previously published values[39] (Fig. 1d). Applying PER-seq to the high-fidelity polymerase KAPA-U⁺ resulted in a 47-fold lower overall error rate compared to that of Klenow-EXO⁻ (Fig. 1d–f). Moreover, PER-seq can be used to measure the directional 'error signature' of each polymerase—the frequency of strand-specific errors (mismatches, that is, nucleotide misincorporation) with respect to the immediate template 5′ and 3′ neighboring bases. For the remainder of the paper, we use the notation 'C:dA' to, for example, denote the misincorporation of A opposite template C. Klenow-EXO⁻ and KAPA-U⁺ polymerases showed distinct error signatures (Fig. 1e–h and Extended Data Fig. 1f), validating that PER-seq can accurately measure both the frequency and sequence specificity of misincorporation by replicative polymerases (Fig. 1e–h).

### Error signature of mutant Pol ε

To elucidate the intrinsic error profile of Pol ε, we first purified the four-subunit (p261, p59, p12 and p17) holocomplex of human Pol ε (wild-type or containing relevant mutations as detailed below) from insect cells using baculovirus expression system[40]. The purified enzyme exhibited DNA polymerase activity, which was determined by the ability of restriction endonucleases to cleave the produced double-stranded DNA (Extended Data Fig. 1 and Supplementary Note 5).

Methylated template reflects the more common physiological state of DNA because around 70% of cytosines in CpGs are methylated in the human genome[41]. We therefore generated a methylated template for PER-seq using M.SssI methyltransferase, which selectively methylates cytosines in a CpG context. Completeness of methylation and the existence of a primed, single-stranded ROI acting as a substrate for Pol ε were confirmed by performing digestions with methylation-sensitive and ROI-recognizing restriction endonucleases (Extended Data Fig. 1a–e). To ensure robustness, experiments were replicated using three different batches of purified Pol ε and two different ROIs.

We performed PER-seq on methylated ROIs with wild-type human Pol ε (PER-POLE-WT), Pol ε containing the P286R mutation in the proofreading domain (PER-POLE-P286R) and Pol ε with a catalytically inactive (D275A/E277A) exonuclease (proofreading) domain (PER-POLE-EXO⁻).

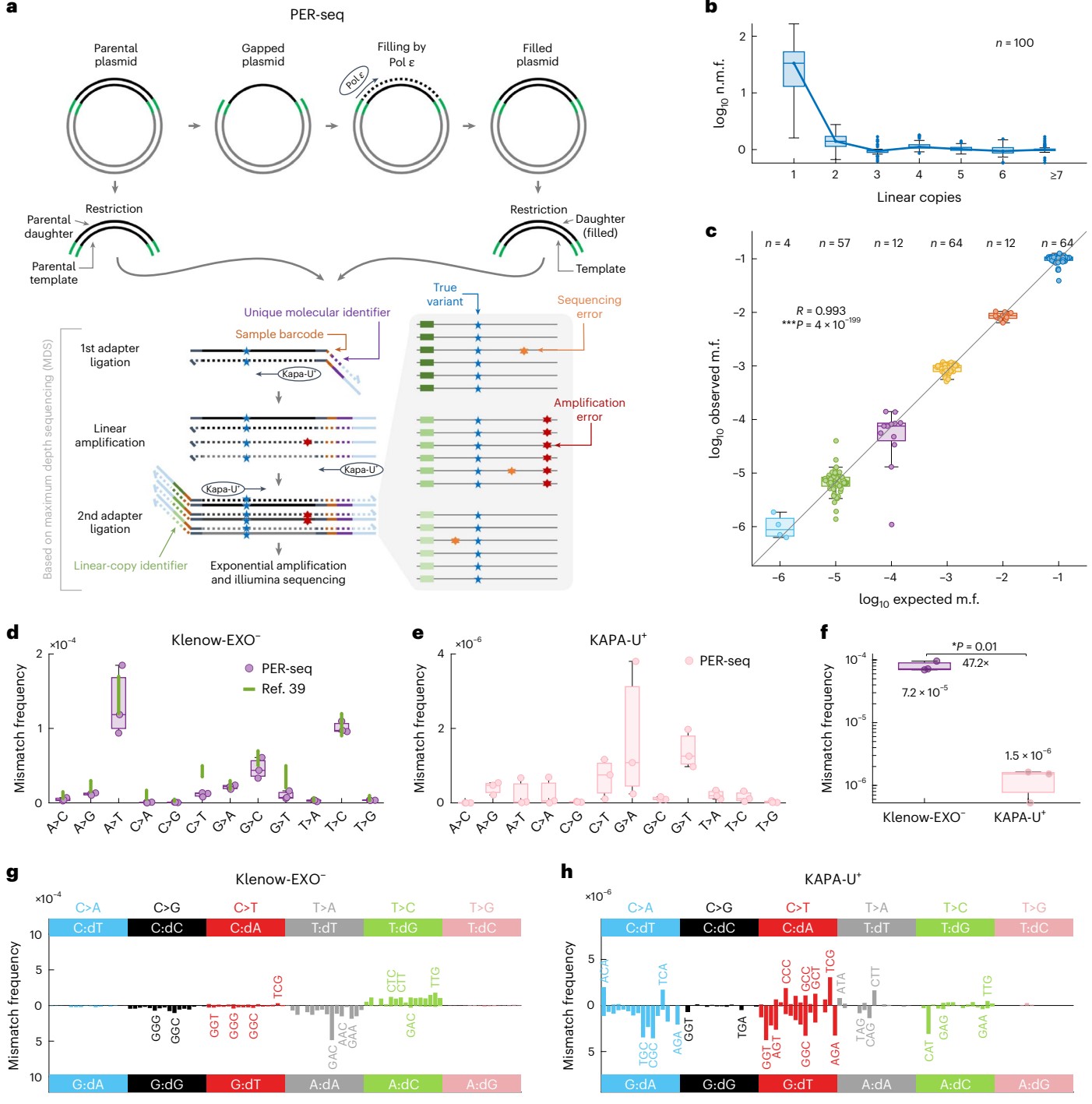

**Fig. 1 | Overview and validation of the PER-seq method. a**, A diagram of the PER-seq method. **b**, Normalized mutation frequency across all samples, shown on a $\log_{10}$ scale, with respect to the required number of linear copies (each with a unique linear-copy identifier). The mutation frequencies were normalized by the average mutation frequency in molecules with at least three linear copies in each sample. **c**, The observed versus expected frequencies of plasmids with artificially introduced mutations spiked in predefined ratios (Methods). Each dot represents one artificial mutant in one sample. Pearson correlation coefficient $R$ and $P$ values are shown. **d**,**e**, Error spectra of individual base changes for Klenow-EXO⁻ (**d**) and KAPA-U⁺ (**e**) measured by PER-seq (after background subtraction and normalization for trinucleotides in the ROI, as in all figures; Methods). $n = 3$

replicates each. The green lines represent the range of previously measured base change error frequencies of Klenow-EXO⁻ (ref. 39). **f**, The average error frequency for Klenow-EXO⁻ and KAPA-U⁺ measured by PER-seq. $P$ values determined by two-sided $t$-test and the ratio of medians are shown. $n = 3$ replicates each. **g**,**h**, Strand-specific error signatures of Klenow-EXO⁻ (**g**) and KAPA-U⁺ (**h**), computed as error (nucleotide misincorporation) spectra with respect to the template 5′ and 3′ neighboring bases (that is, the template trinucleotide), measured by PER-seq and averaged across three replicates. For example, T:dG denotes the misincorporation of guanine opposite thymine on the template strand. Boxplots are plotted with the MATLAB function boxchart (Methods). n.m.f., normalized mutation frequency; m.f., mutation frequencies.

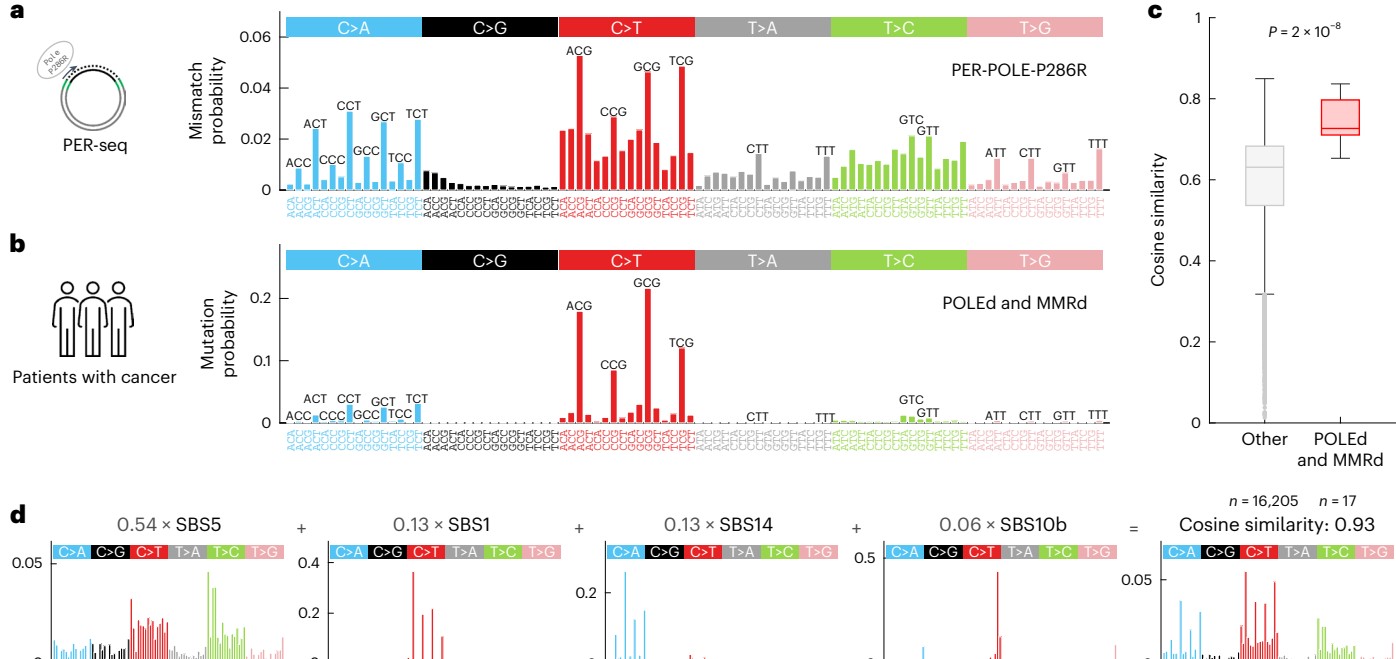

**Fig. 2 | The PER-seq measured error signature of Pol ε P286R resembles the mutational spectrum and mutational signatures of POLEd and MMRd human cancers. a**, The average cell-free PER-POLE-P286R error signature measured by PER-seq and scaled as a probability density function (PDF) to sum to one. All CpGs in the template DNA were methylated. **b**, The average spectrum of mutations in 17 patients with cancer with a combination of a pathogenic mutation in the POLE proofreading domain and a defect in the MMR pathway (POLEd and MMRd cancers), normalized for trinucleotide frequency and scaled as a PDF in the same way as in **a**. **c**, A distribution of the cosine similarity between mutational spectra of human cancer samples to the PER-POLE-P286R error signature shown in **a** (both scaled as a PDF). The red boxplot shows cosine similarity values for POLEd and MMRd cancers, and the gray boxplot shows cosine similarity values for all other cancers. *P* value determined by two-sided, two-sample Mann–Whitney *U* test. **d**, A reconstruction of the PER-POLE-P286R error signature by SBS mutational signatures of the COSMIC-V3 database, using non-negative least square regression (Methods). The linear coefficients for each of the four SBS signatures are shown in gray. The last graph in **d** shows the reconstructed vector (computed as a linear combination of the four SBS signatures) and the resulting cosine similarity to the original PER-POLE-P286R error signature. Boxplots are plotted with the MATLAB function boxchart (Methods).

We initially focused on POLE-P286R because it is the most common pathogenic *POLE* mutation observed in human cancers, and mutational patterns resulting from mutated enzymes have been analyzed before[27]. Our measurements showed that POLE-P286R has a high median error rate of $342 \times 10^{-6}$ per bp and a consistent error signature across the two ROIs and four replicates in total (median pairwise cosine similarity of 0.97; Extended Data Fig. 2a). The average in vitro POLE-P286R (PER-POLE-P286R) error signature after subtraction of assay-specific background is shown in Fig. 2a.

To examine the similarity of our PER-seq measurements to mutations in patients with cancer, we compared the PER-POLE-P286R error signature to the mutational profiles of over 16,000 cancer samples (comprising 13,408 whole-exome and 2,804 whole-genome sequencing (WGS) samples) from the International Cancer Genome Consortium (ICGC) database and other sources[24,42]. The PER-POLE-P286R error signature most closely resembles the mutational profile of a group of 'POLEd and MMRd' samples of patients with cancer that have a pathogenic mutation in the POLE proofreading domain and defects in MMR, the major postreplicative DNA repair pathway (Mann–Whitney *U* test of cosine similarities between groups: $P = 2 \times 10^{-8}$; Fig. 2a–c). In particular, the major peaks in the PER-POLE-P286R signature (CpT>ApT and CpG>TpG) clearly match the major peaks in the POLEd and MMRd average profiles. The PER-POLE-P286R error signature best corresponds to profiles of cancer samples where MMR loss precedes the acquisition of the *POLE* mutation (Supplementary Note 6). Interestingly, the PER-seq measurements recapitulate also the less pronounced but very characteristic TpT>GpT peaks commonly found in the POLEd and MMRd samples (Extended Data Fig. 2b,c).

We used non-negative least squares regression to decompose the PER-POLE-P286R error signature into SBS mutational signatures of the COSMIC-V3 database (https://cancer.sanger.ac.uk/signatures/), determining the subset of signatures that optimally reconstruct the PER-POLE-P286R profile (Methods). The PER-POLE-P286R error signature is best explained by a combination of the following four SBS signatures: widespread signatures SBS1 and SBS5, POLEd-specific signature SBS10b and a POLEd and MMRd-specific signature SBS14 (characterized by CpT>ApT), resulting in a cosine similarity of 0.93 to the PER-POLE-P286R error signature (Fig. 2d and Extended Data Fig. 2d). Together, these observations confirm that our cell-free PER-seq measurements of misincorporation rates closely recapitulate mutational signatures observed in patients with cancer.

In line with our hypothesis, these observations also demonstrate that POLE-P286R has an increased intrinsic propensity to insert adenine opposite template 5mC (5mC:dA), which would lead to CpG>TpG mutations if unresolved. To examine whether the detected misincorporation signature and increased error rate at 5mC are the result of a gain of function specific to the P286R mutation, we compared PER-POLE-P286R with the error signature of the exonuclease-deficient enzyme (PER-POLE-EXO⁻). The absolute error rate of PER-POLE-P286R was 2.2-fold higher than that of PER-POLE-EXO⁻ ($P = 0.001$; Fig. 3a), in line with previous yeast and mouse in vivo functional studies that supported a gain-of-function mutator phenotype of POLE-P286R[18,20,21,23,43]. Notably, PER-POLE-EXO⁻ showed a very similar error profile to PER-POLE-P286R (median pairwise cosine similarity of 0.93; Extended Data Fig. 2a and Supplementary Notes 7 and 8), indicating that the

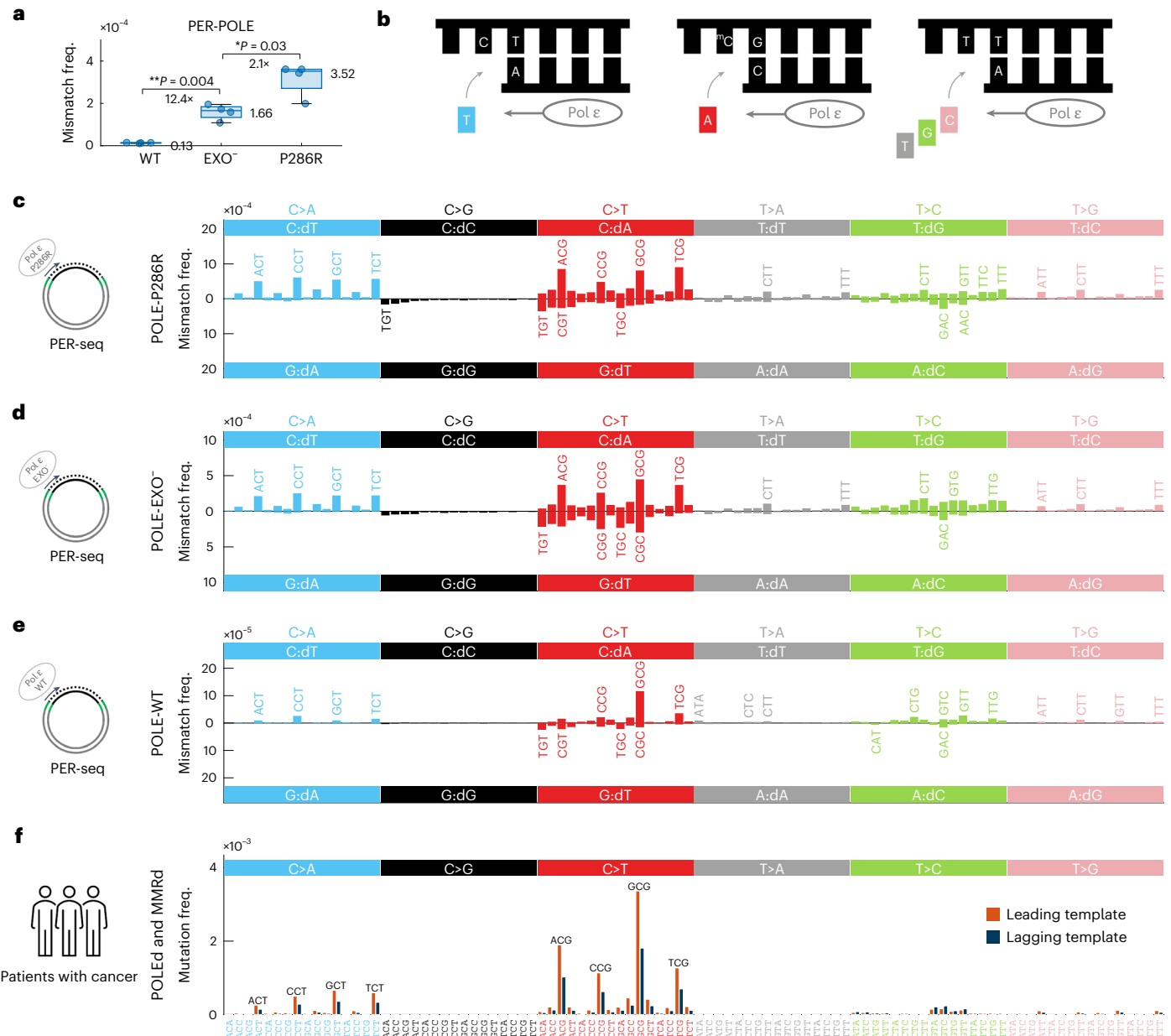

**Fig. 3 | A comparison of POLE-P286R, exonuclease-deficient Pol ε and wild-type Pol ε error spectra determined by PER-seq. a**, The average error frequency for the three polymerases (wild-type (WT), exonuclease-deficience (EXO⁻) and P286R mutant) measured by PER-seq. *P* values determined by paired two-sided *t*-test and the ratio of medians are shown. All CpGs in the template DNA were methylated. *n* = 4 replicates each. **b**, A diagram of the most common misincorporations by Pol ε. The top strand represents the DNA template, and the bottom strand is filled by Pol ε. The red boxes represent the base that is incorrectly incorporated by Pol ε. **c–e**, Strand-specific error signatures of P286R (**c**), EXO⁻ (**d**) and wild-type (**e**) polymerases, computed as error (nucleotide misincorporation) spectra with respect to the template 5′ and 3′ neighboring bases, measured by PER-seq and averaged across four samples. **f**, Average mutation frequency observed in WGS data of POLEd and MMRd human cancers in the leading (dark blue) and lagging (orange) replication strand templates, normalized for trinucleotides in the two strands. Boxplots are plotted with the MATLAB function boxchart (Methods).

increased 5mCpG>TpG error rate of the P286R mutant is an intrinsic feature of the polymerase domain.

Conventionally, mutational signatures are reported from the perspective of the pyrimidine because the strand of the DNA damage is usually unknown. In contrast, PER-seq enables us to distinguish errors when replicating C or T on the template from errors when replicating G or A (Fig. 3b–e). We show that Pol ε intrinsically makes the following three types of errors that depend on the 3′ base of the template: (1) misincorporation of T opposite C in a CpT context, leading to CpT>ApT mutations; (2) A opposite 5mC in a CpG context, and to a lesser extent also T opposite G in a 5mCpGpN context,

both leading to CpG>TpG; and (3) C/G/T opposite T in a TpT context, leading to TpT>(G/C/A)pT.

Next, we aimed to further dissect how Pol ε-induced errors contribute to mutagenesis in patients with cancer. As previously shown by us[13,44] and others[45,46], it is possible to distinguish leading and lagging replication strand errors in cancer somatic mutation data by incorporating information about the direction of DNA replication. Applying this approach to POLEd and MMRd cancer samples, we detected an enrichment of our PER-seq-derived strand-specific errors on the template of the 'leading strand' (Fig. 3f), consistent with the major role of Pol ε in the synthesis of the leading strand[30,47].

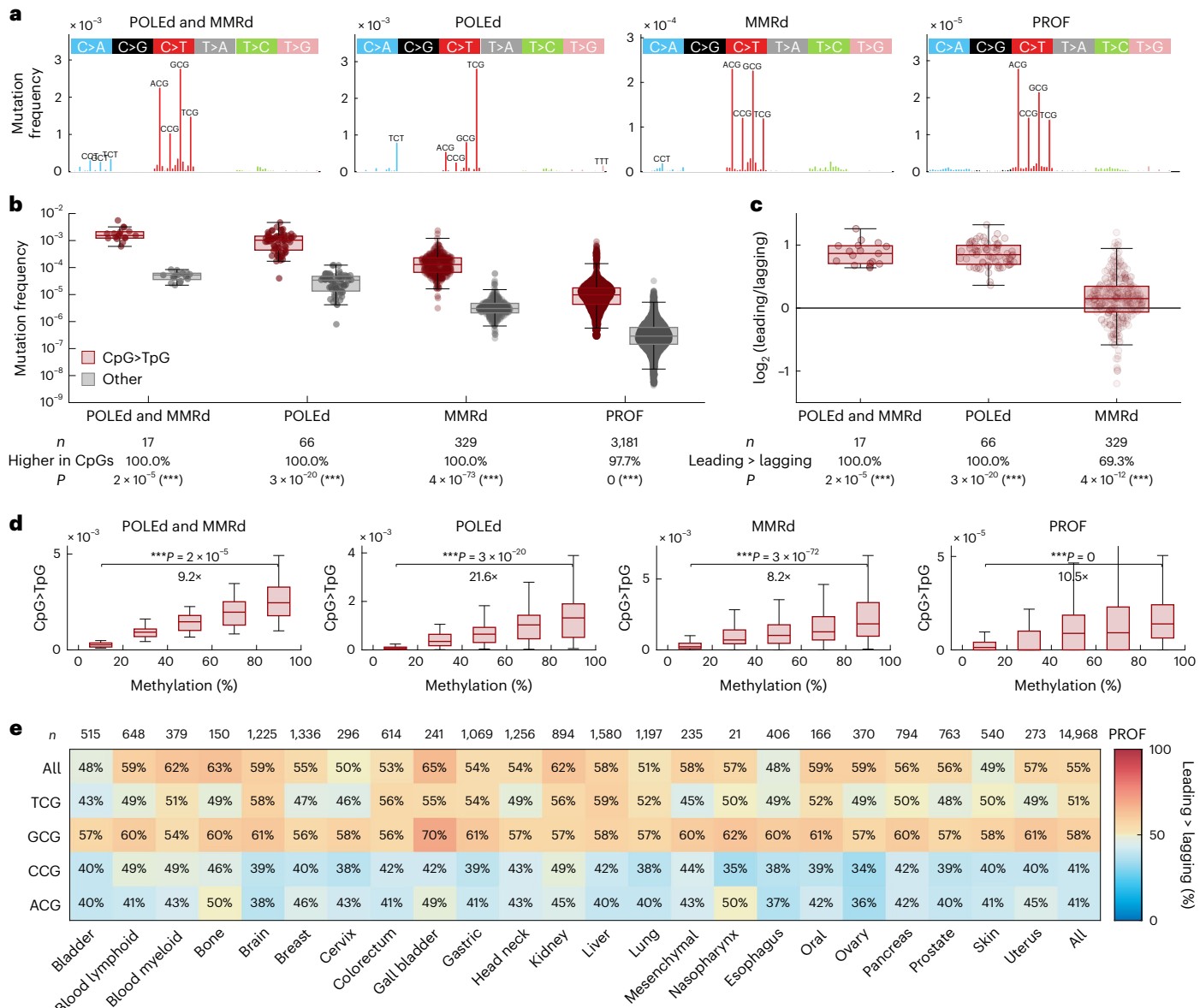

**Fig. 4 | Mutational spectra of POLEd and/or MMRd human cancers support the involvement of replication errors in CpG>TpG mutagenesis. a,** Average mutational spectra in POLEd and MMRd, POLEd (and MMRp), MMRd (and POLEp) and PROF (=POLEp and MMRp) human cancer samples. **b,** Distribution of frequency of CpG>TpG mutations (dark red, per CpG) compared to other mutation types (gray, average frequency of the other 92 mutation types, normalized for trinucleotide occurrences) in these four groups of cancer samples. *P* values determined by two-sided sign test are shown; *P* values rounded to 0 if $P < 5 \times 10^{-324}$. **c,** A $\log_2$ transformation of the ratio of CpG>TpG mutation frequency in the leading and lagging strands. High values represent enrichment on the leading-strand template. *P* values determined by two-sided sign test are shown. **d,** CpG>TpG mutation frequency in CpGs binned by their 5mC levels,

measured by bisulfite sequencing in a matched tissue of origin. The data points in each boxplot represent samples in each group (*n* as in **b**). **e,** Percentage of samples with CpG>TpG mutation frequency higher on the leading strand than the lagging strand, stratified by cancer tissue (columns) and sequence context (rows), with the first row representing all CpGs grouped together. Red values represent higher CpG>TpG frequency on the leading-strand template, and blue values represent higher CpG>TpG frequency on the lagging strand template. To allow comparison of WES and WGS data, analyses in **a**–**e** were restricted to exonic regions only. To make the comparisons tissue adjusted, PROF graphs in **a**–**d** are restricted to the tissue types that contain POLEd and/or MMRd samples (colon/rectum, gastric, uterus and brain); all tissue types are shown in **e**. Boxplots are plotted with the MATLAB function boxchart (Methods).

In particular, we observed that these cancer samples are not only characterized by an extremely high frequency of CpG>TpG mutations (Fig. 4a,b) but also by an enrichment of CpG>TpG mutations on the leading-strand template (Fig. 4c), in line with the replication-linked and deamination-independent origin of these mutations. Moreover, loci with higher 5mC (tissue-matched) also exhibit significantly elevated CpG>TpG mutation frequency (Fig. 4d and Extended Data Fig. 2e). These results are reproducible in exomes (Fig. 4d), whole genomes (Extended Data Fig. 3) and outside exomes (Extended Data

Fig. 4). Together, our combined cell-free and cancer-patient findings demonstrate that 5mC is replicated with decreased fidelity by Pol ε, explaining the high CpG>TpG mutagenesis in POLEd cancer samples (Fig. 4e).

**POLE-P286R causes CpG>TpG mutations in cells and in vivo**

To rule out that our observations are the result of cancer-specific adaptations, we asked whether the high CpG>TpG mutation burden can be reproduced in an engineered cell line and a mouse model of mutant

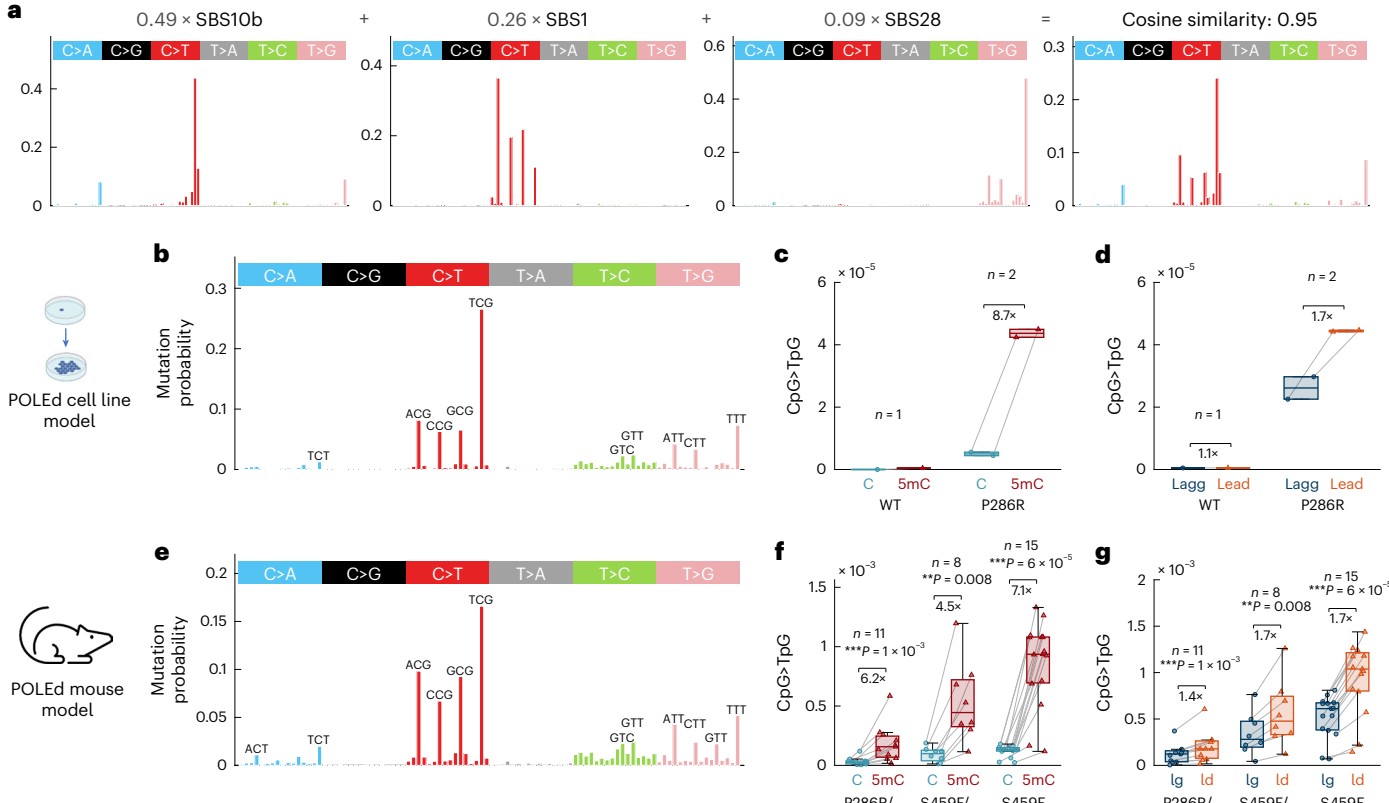

**Fig. 5 | Mutant Pol ε causes CpG>TpG mutations in vitro and in vivo.**
**a**, A reconstruction of the mutational profile of the P286R mutation in mES cells by SBS mutational signatures of the COSMIC-V3 database, using non-negative least square regression. The linear coefficients for each of the four SBS signatures are shown in gray. The last graph in **a** shows the reconstructed vector (computed as a linear combination of the four SBS signatures) and the resulting cosine similarity to the original mES cell P286R mutational profile.
**b**, Normalized mutational profile from WGS of mES cell POLE-P286R clones after 2 months of mutation accumulation and single-cell bottlenecking, averaged across two samples. **c**, CpG>TpG mutation frequency in the mES cell clones (WT versus P286R) in lowly (<20%) and highly (>80%) methylated CpGs, determined

from whole-genome bisulfite sequencing of E14 mES cells (GEO GSM4818066).
**d**, CpG>TpG mutation frequency in the mES cell clones in the lagging and leading strand, estimated from mouse replication timing data. **e**, Normalized mutational profile from tumor WES from CRISPR–Cas9 knock-in germline POLE-P286R or S459F mouse models[18], averaged across 34 samples. **f**, CpG>TpG mutation frequency in the mouse tumors (P286R versus S459F versus S459F/−) in lowly (<20%) and highly (>80%) methylated CpGs, determined from whole-genome bisulfite sequencing of mouse thymus (ENCODE ENCFF850HBL). **g**, CpG>TpG mutation frequency in the mouse tumors in the lagging and leading strand. Boxplots are plotted with the MATLAB function boxchart (Methods). *P* values were determined by two-sided sign test.

Pol ε. First, we used CRISPR–Cas9-facilitated homologous recombination to introduce the P286R mutation in mouse embryonic stem (mES) cells (Methods and Extended Data Fig. 5). We obtained two homozygous mutant cell lines (POLE^P286R) and one wild-type cell line, which underwent the same manipulation. Sequencing of single-cell-derived clones enabled the detection of approximately 13,000 unique de novo mutations in POLE^P286R clones as compared to only 400 mutations in the wild-type clone. The POLE^P286R clones showed high similarity to the PER-POLE-P286R error signature, including very high CpG>TpG burden (SBS1), C>A mutations in a TCT context (SBS10b) and T>G mutations in an NTT context (SBS28; Fig. 5a,b). The CpG>TpG mutations represented the most frequent mutation type in the POLE^P286R clones and exhibited over eightfold enrichment at methylated CpGs compared to unmethylated CpGs (based on mouse bisulfite-sequencing measurements), supporting reduced fidelity of Pol ε when replicating 5mC (Fig. 5c). These mutations were enriched on the leading-strand template, in line with the dominant role of Pol ε in leading-strand synthesis (Fig. 5d).

Second, we analyzed existing whole-exome-sequencing (WES) data from mice with CRISPR–Cas9-mediated knock-in germline P286R and S459F mutations[18]. The observed mutational profile again showed a striking resemblance to the PER-POLE-P286R error signature (cosine similarity 0.8) with high CpG>TpG burden, as well as enrichment in

methylated CpGs and on the leading-strand template (Fig. 5e–g). While the absolute mutation burden differed between the three genotypes (POLE^P286R/+, POLE^S459F/+ and POLE^S459F), the high CpG>TpG rate and enrichment in methylated CpGs and on the leading-strand template were consistent across all three genotypes.

Nuclear extracts from the engineered mES cell POLE^P286R and human HCC2998 cells (naturally POLE^P286R/+) replicated template CpGs with elevated error rates, producing C:dA mismatches (Extended Data Fig. 6). This experiment demonstrates that endogenously produced enzymes together with multiple accessory proteins participating in replication produce elevated numbers of errors when replicating methylated CpGs.

In summary, our results show that mutant Pol ε generates CpG>TpG errors in a pure cell-free setup, nuclear extracts, cell lines, mouse tumors and patients with cancer.

### Error signature of wild-type Pol ε

Having established the impact of mutant Pol ε errors, next we interrogated the error patterns of wild-type Pol ε (PER-POLE-WT). The mutational signature of wild-type polymerase is characterized by similar features as those of the two mutant polymerases, albeit at a 12.4-fold lower overall error rate compared to PER-POLE-EXO⁻ ($P = 7 \times 10^{-6}$; Fig. 3a). Nevertheless, the CpG>TpG error rate of wild-type polymerase

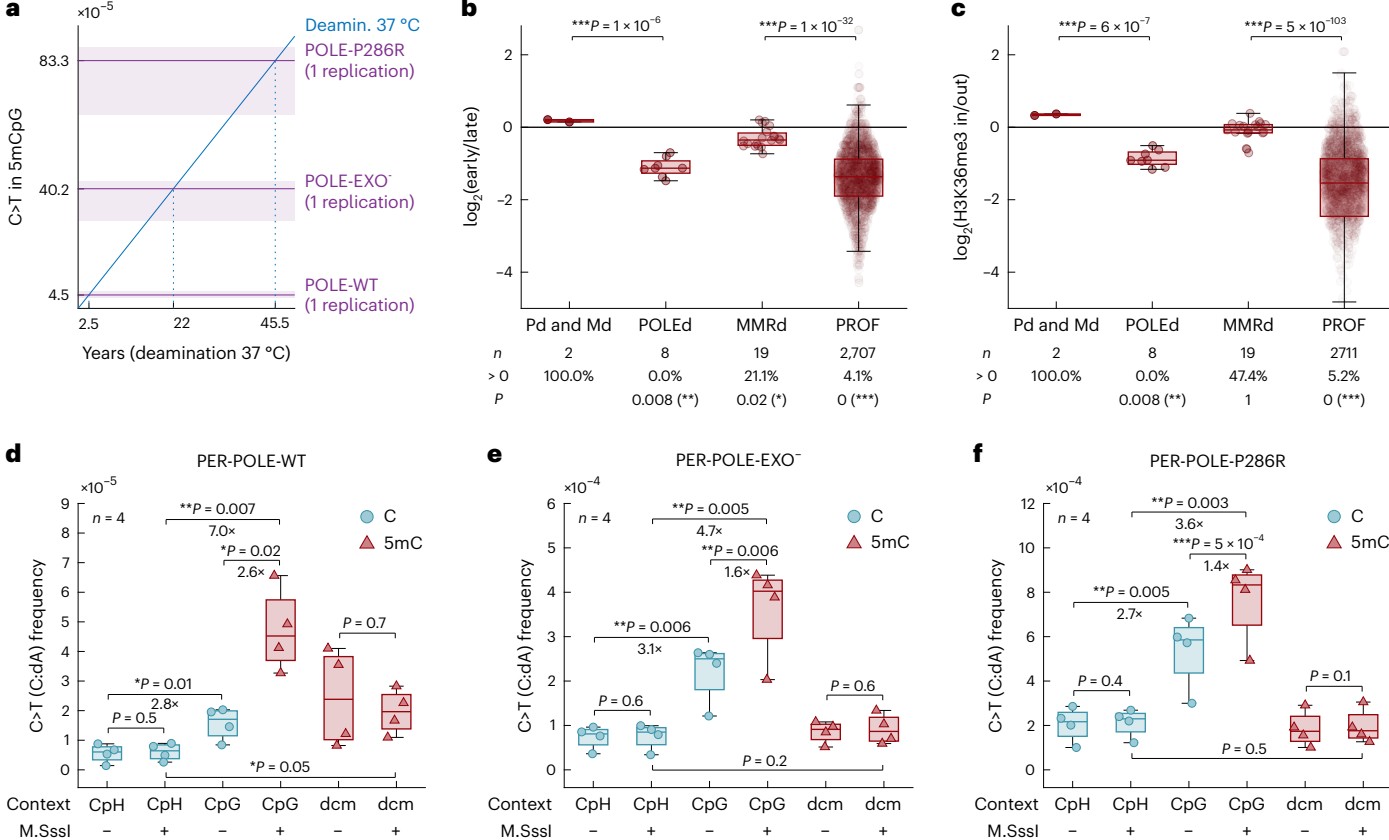

**Fig. 6 | Origins of elevated CpG>TpG mutability. a**, A comparison of the PER-seq measured CpG>TpG error rate in 5mC per single round of replication (purple color) versus previously published estimates of in vitro spontaneous deamination rate of 5mC in double-stranded DNA at 37 °C ($5.8 × 10^{-13}$ per second)[11] (blue color). The *x* axis shows the estimated length of incubation at 37 °C that would generate the same number of CpG>TpG errors as a single round of replication by Pol ε (WT, exonuclease-deficient or P286R). The *y* axis shows the resulting frequency of 5mCpG>TpG errors. **b,c**, CpG>TpG mutations are depleted in MMR-active (early replicating (**b**) or H3K36me3-enriched (**c**)) regions in MMRp but not/less so in MMRd WGS samples. The *y* axis shows a $\log_2$-transformed ratio of CpG>TpG mutation frequency in early/late (**b**) and inside/outside H3K36me3-marked (**c**) regions. Two-sided sign test *P* values (shown below each boxplot) were used to to evaluate whether the values differ from zero. *P* values

comparing samples (shown above each boxplot) were determined by two-sided *t*-test with an uneven variance. **d**–**f**, The PER-seq measured C>T (C:dA) error rate with respect to the modification state and cytosine sequence contexts—CpG, dcm (C**C**AGG and C**C**TGG) and CpH (all other C contexts). Every dot represents the average error frequency in the given context in one sample. Samples with all CpGs methylated by the M.SssI DNA methyltransferase are shown with the plus sign in the bottom row. The color of the boxplots highlights whether the cytosine is methylated (5mC, dark red) or unmodified (C, teal) in the given sample and sequence context. Note that M.SssI presence does not change modification state in CpH or dcm contexts due to its selectivity to CpGs. A paired two-sided *t*-test was used to compare the values between the groups, and the ratio of the medians is shown below the significant *P* values. Boxplots are plotted with the MATLAB function boxchart (Methods).

is substantial ($4.52 × 10^{-6}$), estimated to generate over 2,400 C:dA errors genome-wide in a single replication (Fig. 6a).

The main distinguishing feature of PER-POLE-WT is a higher rate of C:dA errors in a GCG context, compared to that of other CpG contexts (Fig. 3c–e), indicating a reduced capacity by the exonuclease domain to repair these errors. Therefore, we examined the leading versus lagging replication strand asymmetry of CpG>TpG in cancer samples with proficient Pol ε and MMR, with respect to the 5′ base sequence context and tumor tissue of origin (Fig. 4e). The increase of CpG>TpG mutations on the leading strand is most prominent in a GCG context, in line with the elevated error rate in the GCG context in our PER-seq measurements of wild-type Pol ε. These observations suggest that polymerase-induced errors in this context have an elevated likelihood to escape repair and contribute to the accumulation of mutations.

Replication errors that escape Pol ε proofreading are mainly repaired by MMR[20]. We thus investigated whether cancer samples show evidence of MMR-repairing CpG>TpG mutations. MMR has been shown to be more effective in early replicated regions[48] and regions marked with H3K36me3 (refs. [49–53]). We, therefore, tested whether the CpG>TpG mutations in MMR-proficient (MMRp) cancer samples are

depleted in these 'MMR-active' regions. To account for potential confounding correlations with 5mC levels, we focused only on methylated CpGs, using tissue-matched methylation data (Methods). Interestingly, MMRp samples indeed show a depletion of CpG>TpG in both types of MMR-active regions (Fig. 6b,c). Moreover, this depletion is significantly reduced in MMRd samples (Fig. 6b,c). Finally, a similar trend is observed also in POLEd and MMRp versus POLEd and MMRd samples (Fig. 6b,c).

Altogether, our combined in vitro and cancer patient genomic data analysis supports the conclusion that Pol ε errors are an important contributor to the ongoing accumulation of CpG>TpG mutations also in wild-type Pol ε cancers.

## Methylation-independent replication errors at CpG sites

Our PER-seq Pol ε results up to here were derived entirely from applying PER-seq to methylated ROIs. To determine whether the elevated CpG>TpG (C:dA in CpG) error rate we observed is due to the presence of 5mC in the template or a result of the CpG sequence context itself, we performed PER-seq on nonmethylated ROIs, using both mutant and wild-type Pol ε. C:dA error rates outside of CpG contexts were very similar in the M.SssI-treated and mock-treated samples (Fig. 6d–f),

as M.SssI methylates only CpG sites. Surprisingly, replication of non-methylated CpGs showed increased rates of C:dA misincorporation compared to other C contexts, suggesting that Pol ε has an elevated propensity to incorporate A opposite C in a CpG context. The presence of methylation further increased this effect (2.6-fold in PER-POLE-WT, 1.6-fold in PER-POLE-EXO[−] and 1.4-fold in PER-POLE-P286R), resulting in an additive effect of methylation and sequence context on the Pol ε error rate at CpG sites (Fig. 6d–f).

Finally, we sought to measure replication errors at 5mC in a non-CpG context. Template plasmids for PER-seq were prepared in *E. coli*, which have an endogenous Dcm methyltransferase (methylates CCAGG and CCTGG contexts). Consequently, Dcm contexts are always methylated in all ROIs, irrespective of M.SssI treatment. We observed similar misincorporation rates at Dcm contexts to those observed at unmethylated cytosine outside CpG contexts, suggesting that DNA methylation potentiates mutability during replication selectively in CpGs (Fig. 6d–f).

## Discussion

CpG>TpG mutations, the most common mutation type in normal and cancer cells, have been commonly attributed to elevated spontaneous deamination of 5mC, a process independent of replication errors. Our findings challenge this long-standing view in multiple ways. We show that methylated CpGs accrue more errors than any other base when replicated by both mutant as well as wild-type human Pol ε, leading to CpG>TpG mutations at methylated cytosines, independently of deamination. Methylation of cytosine at the 5′ position makes it structurally more similar to thymine, which also features a 5′ methyl group on the pyrimidine ring. This structural similarity could explain the increased probability of misincorporation of A opposite 5mC. Interestingly, our data also show that replication errors are more frequent in CpG contexts even when unmethylated, albeit at a lower rate compared to methylated CpG. Together with the observation that methylation outside of CpG contexts (bacterial dcm sites) does not increase replication errors to the same extent as CpG methylation, this points to a model where both base context and cytosine modification influence Pol ε error rate.

The contribution of replication errors to the generation of CpG>TpG mutations resolves a number of puzzling observations in the data of patients with cancer. First, it explains why patients with cancer with defective Pol ε proofreading or postreplication repair exhibit a disproportionally high CpG>TpG frequency compared to other mutation types[13,54] (Fig. 4a,b). Second, it agrees with the enrichment of CpG>TpG mutations in POLEd samples on the leading strand of replication, which is primarily synthesized by Pol ε (Fig. 4c). Third, it is in line with the correlation between CpG>TpG mutation frequency and 5mC levels in both proofreading-proficient/MMR-proficient and proofreading-deficient/MMR-deficient cancers (Fig. 4d). Fourth, it clarifies why CpG>TpG mutations not only correlate with age but also accumulate more rapidly in fast-replicating tissues compared to tissues with a low turnover rate[5,55,56]. Fifth, it offers an explanation for the correlation of SBS1 with InDel (ID) signatures ID1 and ID2 that are thought to result from slippage at poly-T repeats during DNA replication[57]. Sixth, it clarifies why CpG>TpG mutations are enriched in regions with lower activity of MMR, such as late-replicating regions, and why this relationship is lost in cancers deficient in MMR (Fig. 6b,c). Finally, the contribution of replication errors to CpG>TpG mutagenesis may have implications beyond cancer and provide a possible explanation for the observed sixfold faster germline CpG>TpG mutation rate in paternal compared to maternal DNA[58], as paternal germ cells undergo more cycles of replication than maternal germ cells.

Spontaneous deamination of 5mC and replication-induced accumulation of CpG>TpG mutations are likely to co-occur in living cells. What are the likely contributions of each of these processes to mutation accumulation? We compared our PER-seq measurements of Pol ε error rates to the previously published estimates of the in vitro

deamination rate of 5mC in double-stranded DNA at 37 °C ($2.6 \times 10^{-13}$ per second for unmodified C versus $5.8 \times 10^{-13}$ per second for 5mC)[11]. Notably, it would require incubation for 2.5 years at 37 °C to generate the same number of CpG>TpG errors as a single round of replication by proofreading-proficient Pol ε (Fig. 6a). These results suggest that in replicating cells, polymerase errors may be a larger source of CpG>TpG mutations than spontaneous deamination[11]. It was previously noted that the estimated spontaneous deamination rate—only two to three deamination events per day in each cell—appears too low to explain the observed high frequency of CpG>TpG mutations[2]. Meanwhile, the steady-state levels of many endogenous and exogenous DNA lesions are between hundreds and several thousand per day per cell[2]. Interestingly, for colon cells with a turnover rate of one replication every 5 days, our data predict up to 4,300 CpG>TpG errors per day per cell, of which 480 are expected to escape proofreading. The estimated number of replication-induced CpG>TpG errors is therefore much more similar to that of other known DNA lesions. It is important to note, however, that very few deamination measurements have been performed in double-stranded DNA at 37 °C[11]. Furthermore, it has not been comprehensively studied how deamination of 5mC is impacted by nucleosome occupancy, the local composition of solutes, localized DNA melting during transcription and replication and other cellular processes. Finally, deamination of 5mC results in a T:G mismatch, while Pol ε errors produce 5mC:A. The efficiency of repair of these different types of mismatches remains to be elucidated.

The implications of our findings extend beyond CpG>TpG mutagenesis, shedding new light on the mechanisms underpinning several SBS signatures. The measured PER-POLE-P286R error signature combines features of SBS1, SBS5, SBS14 and SBS10. SBS14 is found specifically in POLEd and MMRd cancer samples[28,46,59]. We experimentally validated in vitro that SBS14 reflects the error signature of human Pol ε in the absence of MMR. A range of putative mechanisms to explain the hypermutation phenotype in POLEd samples have been previously proposed—a simple loss of proofreading[43], expansion of deoxynucleotide triphosphate (dNTP) pools[60], recruitment of error-prone TLS polymerases Pol κ and Pol η[23], involvement of oxidative damage[61–64] and others[43]. The fact that Pol ε P286R in a cell-free environment recreates the characteristic mutational pattern of POLEd and MMRd cancer samples demonstrates that these mutations reflect the intrinsic error signature of Pol ε, independently of any additional factors, such as DNA damage, recruitment of other polymerases or accessory proteins (see Supplementary Note 9 for further discussion, including potential species-specific differences). Conversely, SBS10 is the canonical signature of POLE deficiency in MMRp cancers. Although it is often referred to as the 'POLE signature', our results show that SBS14 better represents the true human 'POLE signature', while SBS10 results from MMR-mediated correction of SBS14, as previously suggested based on observations in yeast[24,60].

The cause of SBS5 is currently unexplained. Our data raise the possibility that polymerase errors are involved in the etiology of SBS5, which would agree with its clock-like properties. In line with this possibility, the highest burden of SBS5 can be observed in patients with POLEd and MMRd cancer; however, future research will be needed to determine whether polymerase errors might underlie SBS5 (Supplementary Note 10).

Our results also shed light on the long-discussed role of Pol ε in leading-strand DNA replication[65,66]. The PER-seq measured error signature of the human Pol ε matches the mutational signatures of POLEd and MMRd cancers, including the directionality of these mutations and their enrichment on the leading strand. Thus, our study confirms the dominant role of Pol ε in leading-strand DNA synthesis.

In summary, we measure the sequence-context-specific misincorporation rate of human wild-type and mutant Pol ε on methylated template DNA, representing the predominant physiological substrate in human cells. We observe an elevated CpG>TpG error rate that is

intrinsic to the polymerase domain and partially escapes proofreading. The resulting mutations likely contribute substantially to the most widespread cancer mutational signature, SBS1. Looking beyond these findings, PER-seq will enable the characterization of the error rate and spectrum of other DNA polymerases and their dependence on environmental conditions such as dNTP ratios and concentrations. The resulting map of replicative fidelity will shed light on the causes of mutation rate variability and could give rise to new cancer-prevention strategies through a reduction of mutational burden.

## Online content

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

## Methods

The presented research complies with relevant ethical regulations. No animal or human studies were conducted that require approval of ethics by responsible authorities.

### DNA polymerases

The nonhuman polymerases used for filling and PER-seq library preparation were KAPA HiFi Uracil+ (Roche, 7959052001), Klenow fragment (NEB, M0212S) and Q5U (NEB, M0515L). The human wild-type and mutant Pol ε were prepared as described previously[40,67]. Briefly, to isolate polymerase complexes, Sf9 cells were coinfected with baculoviruses containing the p261 subunit (either the wild type or mutated), Flag-tagged p59 subunit, p12 subunit and p17 subunit. The enzymes were purified using MonoQ and Flag affinity chromatography together with a final glycerol gradient step[40]. The purity of the polymerase enzymes was assessed after the glycerol gradient step by SDS–PAGE and Coomassie gel staining. Protein concentrations were calculated using serial dilutions of Biorad protein markers. Specific enzyme activity was evaluated for each purification batch to ensure consistent enzyme quality (Supplementary Note 5).

### PER-seq: plasmid preparation

Detailed PER-seq template preparation procedures are described in Supplementary Note 11. In brief, two ROIs from the human genome, exons of *TP53* and *DNMT1* genes, were selected and cloned into the pUC19 vector. Plasmids were methylated with M.SssI (Thermo Fisher Scientific) methyltransferase, and DNA with damaged bases was removed by treatment with uracil-DNA glycosylase (UDG) and formamidopyrimidine-DNA glycosylase (FPG) followed by treatment with T5 exonuclease. A single-stranded gap was introduced by double nicking with Nt.BpU10I (Thermo Fisher Scientific), competitive hybridization to complementary oligonucleotide and size selection.

### PER-seq: filling

The filling with human wild-type and mutant Pol ε was carried out in 100 μl of a buffer containing 20 mM Tris (pH 7.5), 10 mM magnesium acetate, 0.1 mM DTT, 150 mg ml$^{-1}$ BSA together with 100 μM of each nucleotide, 40 fM of polymerase and 40 ng of the indicated DNA template. Reactions were incubated at 37 °C for 30 min. These are based on standard conditions originally set up by the Hurwitz Lab[40].

Filling reactions using bacterial polymerases were performed in 25 μl using 100 ng of gapped plasmid. For KAPA-U$^+$, a 2× mix including dNTPs (Roche, 7959052001) was used, and the reaction was incubated at 72 °C for 3 min. Klenow fragment (NEB, M0212S) filling conditions were 1× NEB2, 1U Klenow fragment and 0.2 mM dNTPs, and elongation was performed at 37 °C for 5 min. Reactions were assembled on ice and transferred back to ice after elongation before magnetic bead purification.

### PER-seq: library preparation

To cut out the ROI, 12 ng of plasmid was digested with 10 U of SacI (NEB) and HindIII (NEB) for 30 min at 37 °C at 225 rpm in a 50 μl reaction before purification on Serapure beads[68] using a volume to bind dsDNA >200 bp and eluted in 9 μl 1 mM Tris–Cl (pH 8.5). In the 'linear 0' step, forward Illumina adapters containing a 19N barcode were annealed to the ROI in a 25 μl reaction containing 0.5U Q5U (NEB; heat-activated before the addition of DNA), 1× Q5U buffer, 200 μM dNTPs, a 20× excess molar ratio (to starting plasmid) of forward adapter (Supplementary Table 1) and either 1.5 ng (for parental samples) or 3 ng (for filled samples) of restricted plasmid. Samples were subjected to a single round of amplification by incubating at 95 °C for 2 s, 55 °C for 1 min and 72 °C for 1 min and were immediately mixed by pipetting with 25 μl master mix containing 2× exonuclease buffer and 20 U exonuclease I (NEB) before incubation at 37 °C for 30 min at 225 rpm to eliminate unused adapter and nontarget strand.

Linear amplification was performed in 45 μl reactions with the P5 Illumina primer (556 nM), dNTPs (222 μM) and Q5U (1 U), heat-activating polymerase separately as before and cycling (95 °C for 5 s, 61 °C for 15 s and 72 °C for 1 min) for seven times. Reactions were then transferred to a fresh PCR tube containing 5 μl 1× Q5U reaction buffer with a 50× excess molar ratio (to starting plasmid) of reverse adapter (Supplementary Table 1) and cycled once using 95 °C for 15 s, 55 °C for 1 min and 72 °C for 1.2 min. Samples were then added to 50 μl of master mix containing 2× exonuclease I reaction buffer and 20 U of exonuclease I (NEB) and incubated at 37 °C for 30 min at 225 rpm. DNA purification was performed in two steps. First, to remove any high molecular weight DNA, samples were added to Serapure beads using a volume to bind dsDNA >700 bp and incubated by rotating for 10 min. The supernatant was then transferred to a fresh tube containing Serapure beads to bind DNA >400 bp, incubated rotating for 10 min, washed 3× in 80% ethanol and eluted in 20 μl 1 mM Tris–Cl (pH 8.5).

Exponential amplification was then performed in a 50 μl reaction including 1 U Q5U and P5 and P7 Illumina primers (500 nM), dNTPs (200 μM), heat-activating Q5U before cycling as before. Five cycles of 95 °C for 15 s, 55 °C for 20 s and 72 °C for 1 min were performed before moving tubes on to ice. To perform the minimum required amount of exponential amplification to each library part before pooling, 2.5 μl of each sample was removed to a fresh 12.5 μl PCR mix identical to the first with the addition of SYBR green to achieve the final concentration of 1× in the 15 μl reaction, while the remainder of the original reaction was kept on ice. Aliquots were then cycled in an ABI StepOne plus QPCR machine and cycled 95 °C for 15 s, 61 °C for 20 s and 72 °C for 1 min for 20 cycles, and the cycle at which each sample passed a predetermined fluorescence threshold was noted and the remainder of the sample cycled in the same conditions for the according number of cycles. Samples were then purified on Serapure beads using conditions that bind dsDNA >400 bp before assessing concentration on a QUBIT and library quality on an agarose gel. Libraries were sequenced on the Illumina NovaSeq platform using 150 bp paired-end sequencing.

### PER-seq: artificial mutants

For quality control and validation purposes, plasmids with defined base substitutions at known positions were spiked in at predefined dilutions. Briefly, site-directed mutagenesis was used to introduce three mutations at each end of the ROI (six in total) in each mutated plasmid (see Supplementary Table 2 for the list of mutated plasmids and used dilutions). Most samples contained three spiked-in plasmids (at dilutions $10^{-1}$, $10^{-3}$ and $10^{-5}$), and selected samples contained seven spiked-in plasmids (at dilutions $10^{-1}$, $10^{-2}$, $10^{-3}$, $10^{-4}$, $2 × 10^{-5}$ and $10^{-6}$).

### PER-seq: data analysis pipeline

Each read in the FASTQ file was split into the sample barcode (usually 6 bp, split into both reads), unique molecular identifier (19 bp, read1), unique linear-copy identifier (4 bp, read2) and the ROI part. Reads were then demultiplexed based on the sample barcode and aligned to the ROI (see details in Supplementary Note 1). Subsequently, reads were grouped by the 'dual barcodes' (combination of unique molecular identifiers (UMI) and unique linear copy identifiers (ULCI)), each representing one linear copy of the molecule. Variant calling was first performed on the level of linear copies, and variants present in at least 70% of reads with the same dual barcode were retained. Finally, the linear copies were further grouped into molecules by their UMI. Only molecules with at least three linear copies (three distinct ULCIs) and variants in at least 70% of linear copies were kept (see Supplementary Notes 1 and 2 for more details and information on the choice of parameters). The summary of all PER-seq samples is shown in Supplementary Table 3.

### PER-seq: background subtraction

To separate true variants (resulting from DNA polymerase misincorporations) from any potential assay-specific artifacts, estimated gapping

background and parental background values were subtracted from the filled daughter strand measurements (see details in Supplementary Notes 2 and 3). This also ensures that observed CpG>TpG mutations are true Pol ε errors and not products of spontaneous deamination (Extended Data Fig. 7 and Supplementary Note 4).

## PER-seq: error signatures

'Mutation/error frequency' was defined as the number of mutations/errors in a given trinucleotide context, divided by the number of occurrences of that context. Strand-specific error signature was then defined as a vector with 192 values, representing the frequency of all three possible alt bases of the error in all 64 possible trinucleotide contexts (4 × 4 × 4). In the strand-specific error signature plots (for example, Figs. 1g–h and 3c–e), pyrimidine changes (that is, C or T on the template strand) are shown in the top part of the plot (upward-facing bars), while purine changes (G or A on the template strand) are shown in the bottom part of the plot (downward-facing bars). Strand-unspecific error signature (for example, Fig. 2a) has 96 values and was computed as the average of the top and bottom parts of the strand-specific error signature (for example, taking the average of C>T in ACG and G>A in CGT), resulting in a signature comparable with the COSMIC SBS signatures[4,57]. The PER-seq measured error signatures of Klenow-EXO⁻, KAPA-U⁺, POLE-P286R, POLE-EXO⁻ and POLE-WT can be found in Supplementary Table 4.

To compute total mutation frequency (across all sequence contexts), the average value of the strand-specific error signature was used. Therefore, all visualizations are normalized for the distribution of trinucleotides in the ROI. For example, in Fig. 1d, e, the frequency of each base change was computed as the average across the 16 trinucleotides shown in Fig. 1g,h. Similarly, the overall mutation frequency shown in Fig. 1f was computed as the average mutation frequency across the 192 values in Fig. 1g,h. In this way, the results are not confounded by the potentially different distribution of trinucleotides in the ROIs.

## Mutation data of patients with cancer

The human cancer data analysis was performed on 13,408 WES and 2,804 WGS samples from the ICGC[69] and other smaller-scale studies[24,42] (Supplementary Table 5). The pan-cancer analysis of whole genomes[70] part of ICGC was used for the WGS samples. For exome-sequencing data, the targetable area was defined by the SureSelectV4 regions (S03723314_Regions.bed) provided by Agilent. In WES, only exome was considered in the analysis, and in both WES and WGS samples, only autosomes were considered. The hg19 reference genome was used throughout this study.

## POLEd and MMRd samples of patients with cancer

POLE-deficient (POLEd) samples were defined as any sample carrying one of the previously published pathogenic exonuclease domain mutations (Supplementary Table 6). MMRd samples were defined as microsatellite instable high (MSI-H, based on The Cancer Genome Atlas (TCGA) FireBrowse MSI test result) or having inherited biallelic mismatch repair deficiency (bMMRd)[24]. POLEd and MMRd samples are those that are both POLEd and MMRd. Seven of 17 POLEd and MMRd samples are bMMRd. Finally, PROF samples were conservatively defined as not carrying any pathogenic exonuclease domain mutations in either POLE or POLD1 (Supplementary Table 6), not MSI (high or low), not bMMRd, not carrying any frameshift/stop-codon mutation in MMR genes (MLH1, MSH2, MSH6 and PMS2) and not having hypermethylation of the MLH1 promoter (mod > 0.25, where available).

## Analysis of mutational spectra

For all cancer samples, the mutational profiles were computed as the frequency of the 96-mutation types (number of mutations in a given trinucleotide context/number of occurrences of the context). For example, the average mutational profile of POLEd and MMRd

samples is shown in Fig. 2b. The similarity of the strand-unspecific PER-POLE-P286R error signature with mutational profiles of individual cancer samples was evaluated using the cosine similarity metric (Fig. 2c). The two-sided Mann–Whitney U test (rank-sum test) was used to compare the cosine similarity values between POLEd and MMRd versus other samples. Boxplots throughout the study are computed and plotted using the box chart MATLAB function and show the median as the center line, the lower and upper quartiles as the box bounds and the minimum and maximum values that are not outliers as the whiskers. Outliers are values that are more than 1.5-fold interquartile range away from the top or bottom of the box.

## Reconstruction by mutational SBS signatures

The v2 and v3.3.1 SBS mutational signatures were downloaded from the COSMIC website. We used non-negative least squares regression to determine the minimal subset of SBS signatures that optimally reconstruct the strand-unspecific PER-POLE-P286R error signature. For a given $K$ = number of SBS signatures, all combinations of $K$ COSMIC SBS signatures were explored, and the combination that leads to the lowest reconstruction error (norm($C \times \mathbf{x} - d$)$^2$, where $C$ is the input matrix of the $K$ signatures, $\mathbf{x}$ is the output vector of exposures and $d$ the input PER-POLE-P286R error signature) was selected. This process was iterated for $K$ = 1, 2, …, and the smallest $K$ that leads to at least a 20% decrease in the reconstruction error was selected, similarly as in the approach in ref. 71 to avoid overfitting.

## Direction of replication

Left- and right-replicating domains were taken from ref. 45 Each domain is 20 kb wide and annotated with the direction of replication and with replication timing. The leading-strand template corresponds to the plus strand in the left direction and the minus strand in the right direction, and vice versa for the lagging strand template. Mutational frequency of the 96-mutation types of WGS POLEd and MMRd samples was computed in the leading and lagging strand templates (Fig. 3f). Finally, the CpG>TpG mutation frequency was compared between the leading and lagging strands in different groups of samples, considering only samples with at least one mutation in one of the two annotated strands (Fig. 4c,e). A two-sided sign test was used to compare the mutation frequency between the two strands.

## DNA methylation

Maps of cytosine modifications (Supplementary Table 7) were obtained from bisulfite sequencing (BS-seq) datasets from the data portals of TCGA, Roadmap Epigenome, Blueprint and from previously published data[72–75]. Coordinates were converted to hg19 using UCSC liftover where necessary. For brain, kidney and prostate maps, raw reads were processed with Trim_galore, Bismark[76] and MarkDuplicates from Picard tools, and only sites covered with at least five reads were considered.

## Introducing P286R mutation in mES cells

P286R mutation was introduced into E14 mES cells using CRISPR–Cas9-assisted homologous recombination as described in detail in Supplementary Note 12 (ref. 77).

## WGS

Genomic DNA (gDNA) was isolated using GeneJet gDNA Purification Kit (Thermo Fisher Scientific) following the manufacturer's protocol. gDNA was fragmented on Covaris S220 using the manufacturer-provided shearing protocol for a target fragment size of 500 bp. gDNA was size selected with 0.55× and 0.3× AMPure XP beads (Beckman Coulter). Libraries were prepared with 100 ng of size-selected DNA using a KAPA HyperPrep PCR-free kit and barcoded with KAPA UDI for Illumina (Roche Diagnostics) according to the manufacturer's protocol. The DNA library was purified with 0.8× AMPure XP beads. Quantification and fragment analysis were performed throughout with Qubit

dsDNA High-Sensitivity Quantification Kit (Invitrogen) and Bioanalyser High-Sensitivity DNA Kit (Agilent) according to the manufacturer's protocol.

## Analysis of mouse WGS data

All executable workflow scripts and R notebooks used in the analysis are available from the code repository linked below. Briefly, paired-end reads were adapter and quality trimmed using TrimGalore, aligned with BWA-MEM to the mm10 genome downloaded from the UCSC genome browser website. Duplicates were marked with MarkDuplicates from the GATK toolset. Variants were called using Octopus (v0.7.0) in 'germline' mode. Variants were considered as 'de novo' if the variant was called in only one sample, the position was sufficiently covered in all samples (between 10 and 40 reads), no other sample showed below-threshold evidence for the variant and the variant allele frequency was between 0.25 and 0.75.

## PER-EXTRACT-seq

Preparation of nuclear extract and template filling was performed as described previously[78] with some minor modifications as explained in Supplementary Note 13.

## Comparison with spontaneous deamination

The PER-seq measured median number of C>T errors by wild-type Pol ε per 5mCpG is as follows:

$$E = 4.5 \times 10^{-5} \text{ errors per replication per 5mCpG.}$$

The previously published estimate of in vitro deamination rate of 5mC in dsDNA at 37 °C is as follows[11]:

$$\text{deaminationRate} = 5.8 \times 10^{-13} \text{ events per second per 5mCpG.}$$

The expected number of deamination events per 5mCpG per year was estimated as follows:

$$R = \text{deaminationRate} \times \text{secondsInAYear} = 5.8 \times 10^{-13} \times 365.2425 \times 24 \times 60 \times 60 = 1.83 \times 10^{-5}.$$

The duration of incubation at 37 °C that would generate the same number of CpG>TpG mutations by spontaneous deamination as a single round of replication by wild-type Pol ε was estimated as follows:

$$D = E/R = 2.5 \text{ years}$$

## Estimates of Pol ε errors per genome per replication

The estimated number of CpG>TpG errors per day per cell due to spontaneous deamination (*E1*), Pol ε before proofreading (*E2*) and Pol ε that escape proofreading (*E3*) was calculated as follows:

$$E1 = \text{deaminationRate} \times \text{secondsInADay} \times \text{nCpGs} = 5.8 \times 10^{-13} \times 24 \times 60 \times 60 \times 53.5 \times 10^6 = 2.68$$
$$E2 = \text{errorPerRepl\_exo} \times \text{replicationsInADay} \times \text{nCpGs} = 40.23 \times 10^{-5} \times 0.2 \times 53.5 \times 10^6 = 4305$$
$$E3 = \text{errorPerRepl\_wt} \times \text{replicationsInADay} \times \text{nCpGs} = 4.52 \times 10^{-5} \times 0.2 \times 53.5 \times 10^6 = 484$$

## MMR-active genomic regions

The replication timing profiles were taken from ref. 45. All CpGs were annotated with the replication timing values, and CpGs in the early and late-replicating regions were defined as the bottom and top quartiles. Tissue-matched H3K36me3 values were obtained as narrowPeak files from ENCODE (Supplementary Table 7). For tissues where H3K36me3 measurements were not available, the consensus (defined as the presence of a peak in at least half of the tissues) was used. This analysis was only restricted to WGS samples and methylated CpGs (defined as tissue-matched BS-seq $\beta$ value of at least 90%) to ensure that the analysis is not confounded by 5mC levels in different genomic regions. Finally, the CpG>TpG mutation frequency was compared between the early versus later-replicated regions and regions inside versus outside H3K4me3 peaks. A two-sided sign test was used to compare the mutation frequency between these groups of CpGs. Two-sample *t*-test with uneven variance was used to compare the $\log_2$ ratios between different groups of samples (POLEd and MMRd versus POLEd, and MMRd versus PROF).

## Statistics and reproducibility

Experiments were reproduced as indicated in all relevant sections and figures. No data were excluded from analyses. No statistical methods were used to predetermine the sample size. The experiments were not randomized. The investigators were not blinded to allocation during experiments and outcome assessment.

## Reporting summary

Further information on research design is available in the Nature Portfolio Reporting Summary linked to this article.

## Data availability

PER-seq sequencing data have been deposited in the Sequence Read Archive under accession SRP439101, and the processed files are available together with the code (see below). The used publicly available cancer samples are listed in Supplementary Table 5. Source data are provided with this paper.

## Code availability

The PER-seq analysis pipeline can be found at https://bitbucket.org/licroxford/per-seq, and it comprises all steps, including sample demultiplexing, trimming based on base quality, mapping, variant calling, calculating corrected mutation frequencies and polymerase error spectra. Analysis of human cancer samples, as well as the code to reproduce the figures and tables from this manuscript, can be found at https://bitbucket.org/licroxford/cpg_mutagenesis. The code is also available at https://doi.org/10.6084/m9.figshare.27089494 (ref. 79). Data analysis was performed in R and MATLAB R2022a and uses the Statistics and Machine Learning Toolbox. The PER-seq pipeline makes use of bedtools (v2.27.0), FastQC (v.0.11.8), Bowtie2 (v.2.3.5.1) and SAMtools (v.1.9).

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

## Acknowledgements

We thank H. Tagoh for advice; C. Goding, R. Klose and B. Lehner for
comments on the manuscript; and A. Green and other team members
in the Oxford Genomic Center for sequencing. This work was funded
by the Ludwig Institute for Cancer Research (to B.S.B. and S.K.), the
Biotechnology and Biological Sciences Research Council (BB/M001873/1
to S.K.) and the Wellcome Trust (225678/Z/22/Z to M.T.). S.K. Lab is
also funded by the National Institute for Health Research (NIHR) Oxford
Biomedical Research Center (BRC) and the Conrad N. Hilton Foundation.
B.S.B. acknowledges funding from the Medical Research Council
(MR/T04490X/1). The results here are in part based on data generated by
the TCGA Research Network (https://www.cancer.gov/tcga).

## Author contributions

M.T., M.M., B.S.B. and S.K. jointly conceived the project and wrote
the manuscript, with contributions from J.T., G.C. and S.C. M.M.,
S.K. and M.T. developed PER-seq, with contributions from B.S.B.
M.M. performed all PER-seq experiments. M.T. developed PER-seq
processing tools and performed all computational analyses, with
contributions from J.T. G.C. and S.C. purified human DNA polymerases
and performed reactions with the enzymes. A.S. and N.M. produced
mES cells with the P286R mutation. E.S. prepared the WGS library
and sequenced the samples. B.S.B. and S.K. acquired funding and
supervised the work.

## Competing interests

The authors declare no competing interests.

## Additional information

**Extended data** is available for this paper at

**Supplementary information** The online version
contains supplementary material available at

**Correspondence and requests for materials** should be addressed
to Marketa Tomkova, Michael John McClellan, Benjamin
Schuster-Böckler or Skirmantas Kriaucionis.

**Peer review information** *Nature Genetics* thanks Ruben van Boxtel,
Radhakrishnan Sabarinathan, Scott Lujan and the other, anonymous,
reviewer(s) for their contribution to the peer review of this work. Peer
reviewer reports are available.

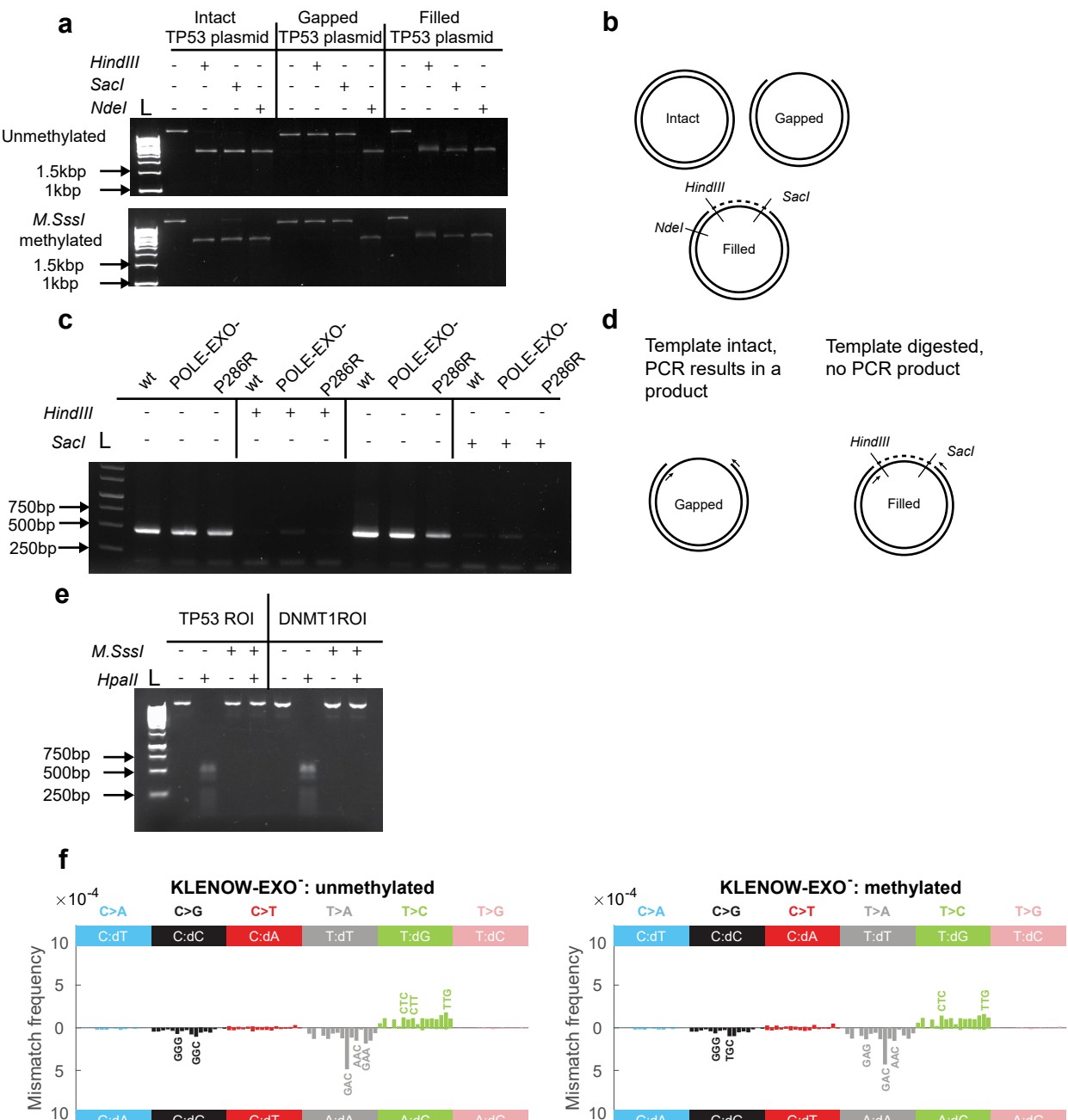

**Extended Data Fig. 1 | Evaluation of DNA polymerase activity and Klenow enzyme error sign. a**, Agarose gels of intact, gapped and filled plasmid following digestion by restriction enzymes within the gapping region (HindIII and SacI) and outside the gapping region (NdeI) for both unmethylated and methylated plasmid. Gapped plasmid is not digested by HindIII or SacI but is linearized by digest of NdeI. 'L' indicates 1 kb GeneRuler (Thermo Fisher Scientific). **b**, Schematic showing the location of restriction sites within the plasmid. Experiments were repeated for each batch of plasmid (N = 2). **c**, Evaluation of template fill-in by DNA polymerases. Templates filled in by indicated polymerases were digested with HindIII or SacI restriction enzymes as shown in **d**. Experiments were repeated for each batch of polymerases (N = 2). **d**, Schematic

showing primer localization sites and restriction enzymes. Single-stranded DNA resists digestion, resulting in the presence of the template for PCR amplification. **e**, Methylation-sensitive restriction enzyme HpaII was used to determine the efficiency of methylation with M.SssI. Experiments were repeated for each batch of plasmid (N = 2). **f**, Strand-specific error signature of Klenow-EXO⁻, when unmethylated and methylated template DNA was used for fill-in. Strand-specific error signatures of Klenow-EXO⁻ and KAPA-U⁺. The error signature is computed as an error (nucleotide misincorporation) spectra with respect to the template 5′ and 3′ neighboring bases (that is, the template trinucleotide), measured by PER-seq and averaged across three replicates. For example, T:dG denotes misincorporation of guanine opposite thymine on the template strand.

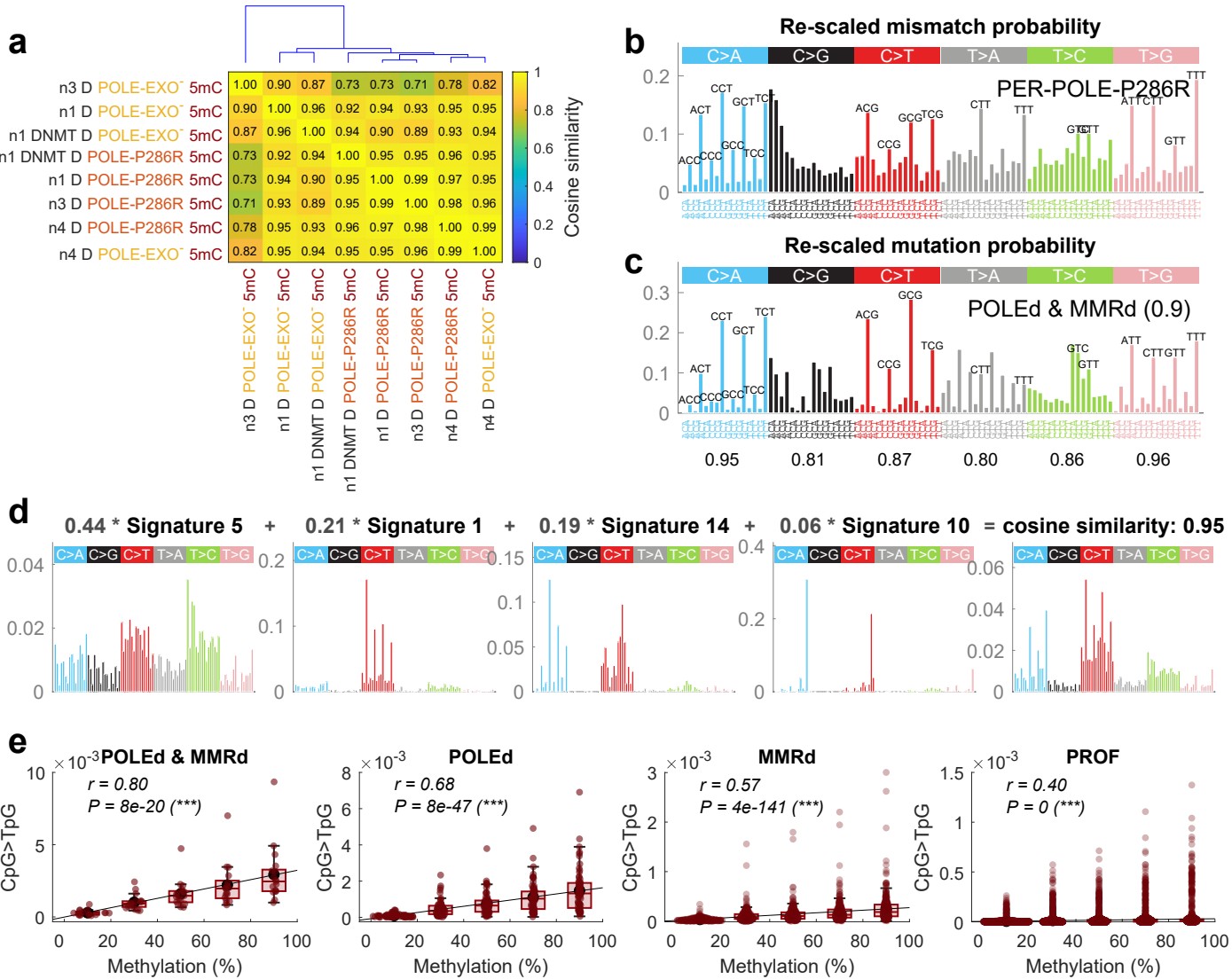

**Extended Data Fig. 2 | The PER-seq measured error signature of Pol ε P286R resembles the mutational spectrum and mutational signatures of POLEd and MMRd human cancers. a**, A heatmap and hierarchical clustering on a pairwise cosine similarity matrix between PER-POLE-P286R and PER-POLE-EXO- samples. The cosine similarity is computed on the strand-specific error spectra (that is, each with 192 error types) after background subtraction and trinucleotide frequency normalization. The hierarchical clustering is computed using the MATLAB functions linkage, optimalleaforder and dendrogram with default parameters. **b,c**, Error/mutational spectra rescaled within each of the six nucleotide substitutions (divided by the sum of all bars of the same color). In other words, this visualization shows the relative mutation frequencies within each nucleotide substitution group. **b**, The average in vitro POLE-P286R ('PER-POLE-P286R') error spectrum measured by PER-seq, after subtraction of assay-specific background, normalized for trinucleotide frequency and scaled as a probability density function in each of the six substitution types. **c**, The average in vivo spectrum of mutations in 17 human cancers with a combination of a pathogenic mutation in the POLE proofreading domain and a defect in the mismatch repair pathway (POLEd and MMRd cancers), normalized for trinucleotide frequency and scaled as a probability density function in each of the six substitution types. The numbers below the profile plot in **c** denote the cosine similarity values between **b** and **c** computed for each of the six substitution types. Interestingly, all six substitution classes exhibit a relatively high cosine similarity, with a minimum of 0.8 in T>A and a maximum of 0.97 in T>G (mainly TpT>GpT). The overall cosine similarity on the rescaled profiles is 0.9. **d**, A reconstruction of the PER-POLE-P286R error signature by SBS mutational signatures of the COSMIC-V2 database, using non-negative least square regression (Methods). The linear coefficients for each of the four SBS signatures are shown in gray. The last panel shows the reconstructed vector (computed as a linear combination of the four SBS signatures) and the resulting cosine similarity to the original PER-POLE-P286R error signature. **e**, CpG>TpG mutation frequency in CpGs binned by their 5mC levels, measured by bisulfite sequencing in a matched tissue of origin. Each dot represents a value in one sample and one 5mC bin (N: 17 for POLEd and MMRd, 66 for POLEd, 329 for MMRd, 3181 for PROF). Spearman correlation coefficient and two-sided P-value are shown on top. Boxplots are plotted with the MATLAB function boxchart (Methods).

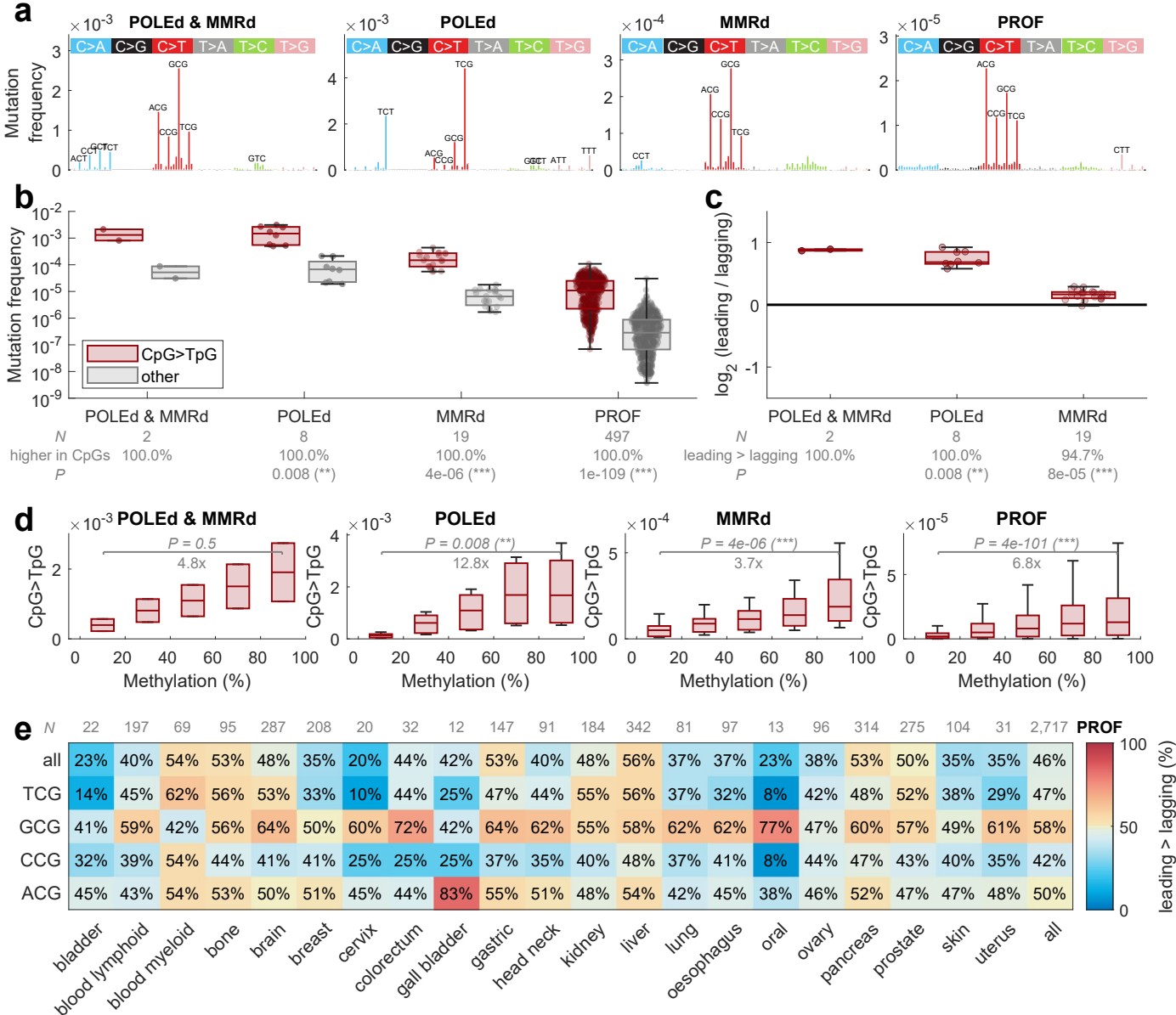

**Extended Data Fig. 3 | CpG>TpG mutagenesis in cancer patients (WGS, entire genome). a**, Average mutational spectra in POLEd and MMRd, POLEd (and MMRp), MMRd (and POLEp) and PROF (=POLEp and MMRp) human cancer samples. **b**, Distribution of frequency of CpG>TpG mutations (dark red, per CpG) compared to other mutation types (gray, average frequency of the other 92 mutation types, normalized for trinucleotide occurrences) in these four groups of cancer samples. The gray text below the boxplots shows 'N': the number of samples, 'higher in CpGs': the percentage of samples with higher CpG>TpG mutation frequency compared to the frequency of other mutation types and 'P': two-sided sign test P-value comparison between the CpG>TpG vs. other mutation frequencies. **c**, A $\log_2$ transformation of the ratio of CpG>TpG mutation frequency in the leading and lagging strands. High values represent enrichment on the leading-strand template. Two-sided sign test P-value is shown in each group. **d**, CpG>TpG mutation frequency in CpGs binned by their 5mC

levels, measured by bisulfite sequencing in a matched tissue of origin. The data points in each boxplot represent samples in each group (N as in **b**). Two-sided sign test P-value is used to compare CpG>TpG frequency between the first and the last bin. **e**, The heatmap color and text represent the percentage of samples with CpG>TpG mutation frequency higher on the leading strand compared to the lagging strand, stratified by cancer tissue (columns) and sequence context (rows), with the first row representing all CpGs grouped together. Red values represent higher CpG>TpG frequency on the leading-strand template, and blue values represent higher CpG>TpG frequency on the lagging strand template. To make the comparisons tissue adjusted, PROF panels in **a**–**d** are restricted to the tissue types that contain POLEd and/or MMRd samples (colon/rectum, gastric, uterus and brain). **e** shows all tissue types. Boxplots are plotted with the MATLAB function boxchart (Methods).

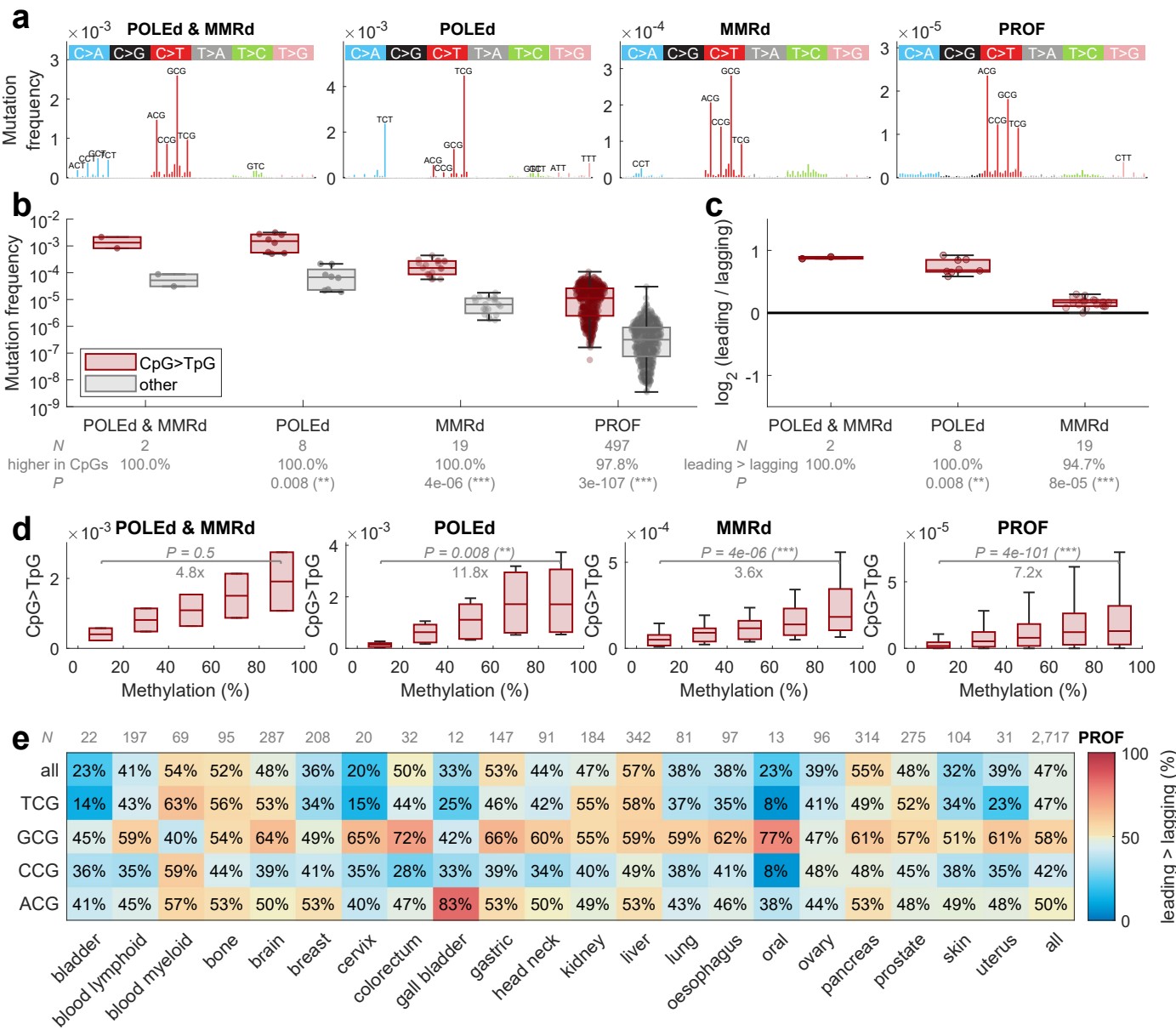

**Extended Data Fig. 4 | CpG>TpG mutagenesis in cancer patients (WGS, outside exome). a**, Average mutational spectra in POLEd and MMRd, POLEd (and MMRp), MMRd (and POLEp) and PROF (=POLEp and MMRp) human cancer samples. **b**, Distribution of frequency of CpG>TpG mutations (dark red, per CpG) compared to other mutation types (gray, average frequency of the other 92 mutation types, normalized for trinucleotide occurrences) in these four groups of cancer samples. The gray text below the boxplots shows 'N': the number of samples, 'higher in CpGs': the percentage of samples with higher CpG>TpG mutation frequency compared to the frequency of other mutation types and 'P': two-sided sign test P-value comparison between the CpG>TpG vs. other mutation frequencies. **c**, A $\log_2$ transformation of the ratio of CpG>TpG mutation frequency in the leading and lagging strands. High values represent enrichment on the leading-strand template. Two-sided sign test P-value is shown in each group. **d**, CpG>TpG mutation frequency in CpGs binned by their 5mC

levels, measured by bisulfite sequencing in a matched tissue of origin. The data points in each boxplot represent samples in each group (N as in **b**). Two-sided sign test P-value is used to compare CpG>TpG frequency between the first and the last bin. **e**, The heatmap color and text represent the percentage of samples with CpG>TpG mutation frequency higher on the leading strand compared to the lagging strand, stratified by cancer tissue (columns) and sequence context (rows), with the first row representing all CpGs grouped together. Red values represent higher CpG>TpG frequency on the leading-strand template, and blue values represent higher CpG>TpG frequency on the lagging strand template. To make the comparisons tissue adjusted, PROF panels in **a**–**d** are restricted to the tissue types that contain POLEd and/or MMRd samples (colon/rectum, gastric, uterus and brain). **e** shows all tissue types. Boxplots are plotted with the MATLAB function boxchart (Methods).

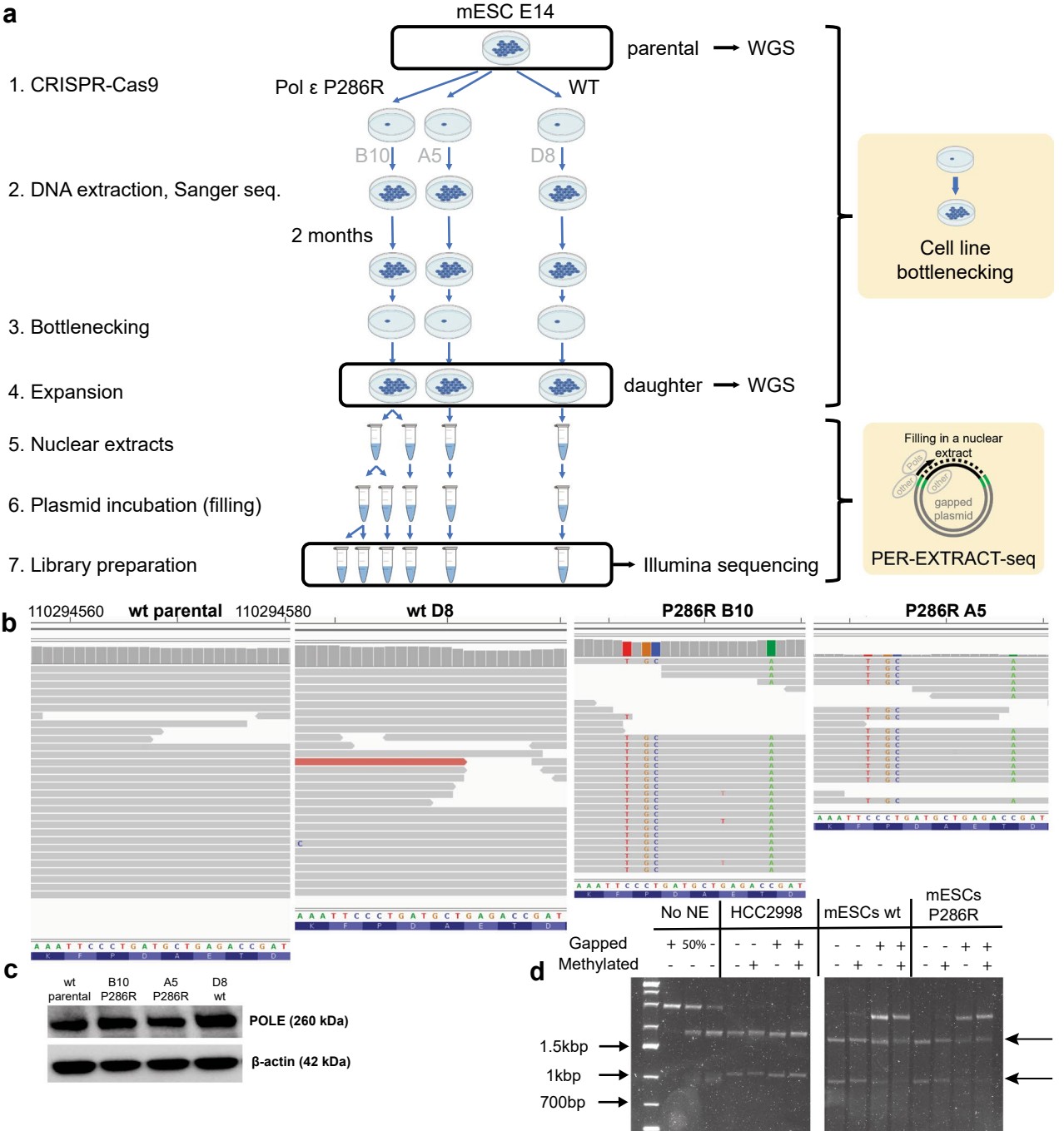

**Extended Data Fig. 5 | WGS of mESCs and PER-EXTRACT-seq. a**, Schematics of experiment for WGS and PER-EXTRACT-seq. **b**, Screenshot from IGV browser displaying reads (horizontal blocks) aligned to mouse genome (Chr5, mm10). Nucleotide variants that do not match annotation are highlighted in the read. B10 and A5 clones have C>G and T>C mutations, which results in P286R mutation and silent C>T (creates BbsI restriction site) and C>A (CRISPR PAM site) mutations. B10 clone also has evidence of unintended G>T mutation in three reads, which

would result in STOP codon in one allele. **c**, Western blot using POLE and β-actin antibodies. Similar level of POLE expression is observed in different clones. **d**, Gapped plasmid (+) resists digestion, and filled plasmid (−) can be digested as shown in lanes containing known amounts of purified DNA (the first three lanes). HCC2998 cell extract completely filled the template, while there was substantial plasmid unfilled in mESCs. As explained in the text, only filled plasmid contributes to the PER-EXTRACT-seq results.

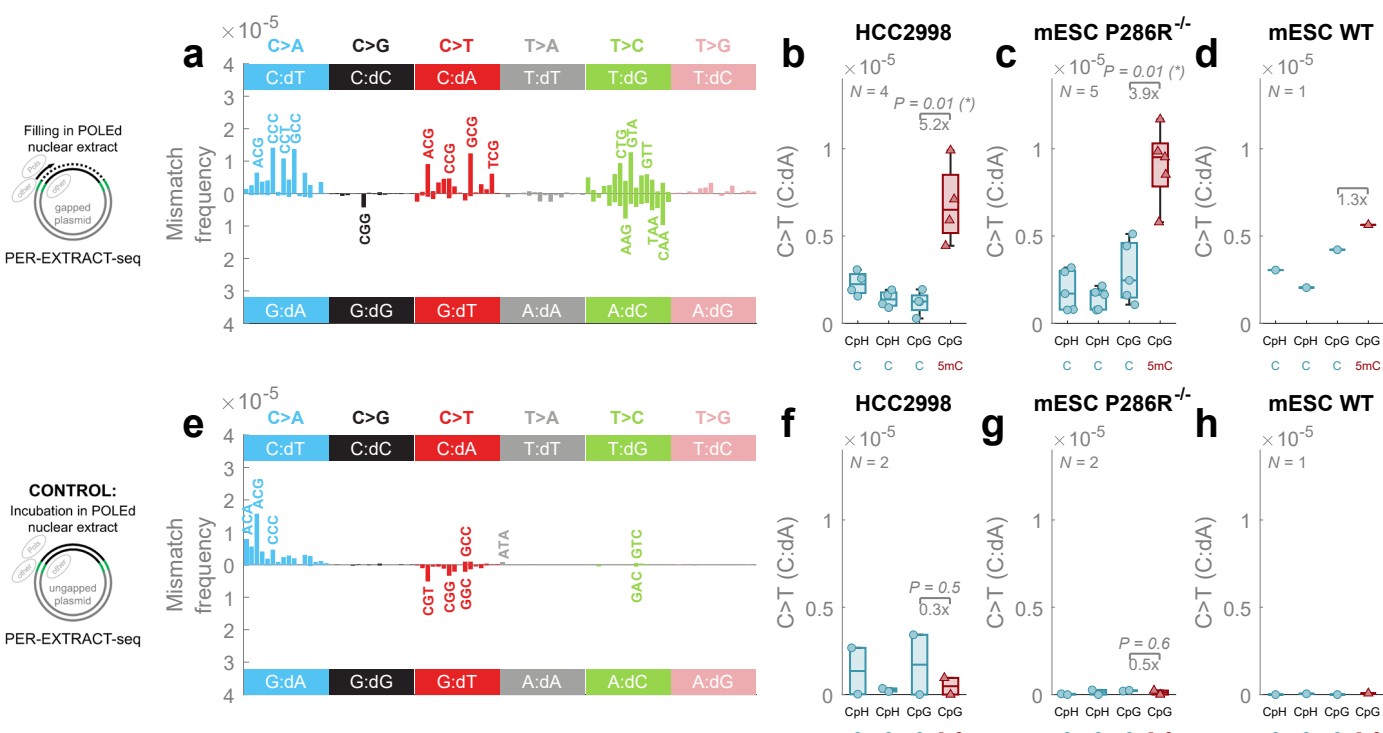

**Extended Data Fig. 6 | PER-EXTRACT-seq results. a**, PER-EXTRACT-seq error signature of filling gapped plasmids in nuclear extracts from cells with POLEP286R. The error signature is computed as error (nucleotide misincorporation) spectra with respect to the template 5′ and 3′ neighboring bases (that is, the template trinucleotide), measured by PER-EXTRACT-seq and averaged across available samples: 5 samples from nuclear extracts from the mESC clones with POLEP286R mutation, and 4 samples from nuclear extracts from HCC2998 cell line (that naturally harbors a POLEP286R/+ mutation). **b–d**, PER-EXTRACT-seq measured C>T (C:dA) error rate with respect to the modification state and cytosine sequence contexts: CpG and CpH (all other C contexts). Every dot represents average error frequency in the given context in one sample. Samples with all CpGs methylated by the M.SssI DNA methyltransferase are shown with the plus sign in the bottom row. The color of the boxplots highlights whether the template cytosine is methylated (5mC, dark

red) or unmodified (C, light blue) in the given sample and sequence context. Note that M.SssI presence does not change modification state in CpH due to its selectivity to CpGs. A two-sided paired t-test was used to compare the values between the groups, and the ratio of the medians is shown below the significant P-values. The values from PER-EXTRACT-seq for filling in HCC2998 (**b**), mESC POLEP286R (**c**) and mESC WT (**d**) nuclear extracts are shown. **e**, PER-EXTRACT-seq error signature of incubating the control ungapped plasmids in nuclear extracts from cells with POLEP286R, averaged across available samples: 5 samples from nuclear extracts from the mESC clones with POLEP286R mutation, and 4 samples from nuclear extracts from HCC2998 cell line. **f–h**, PER-EXTRACT-seq measured C>T (C:dA) error rate in the control ungapped plasmids. A two-sided paired t-test was used to compare the values between the groups, and the ratio of the medians is shown below the significant P-values. Boxplots are plotted with the MATLAB function boxchart (Methods).

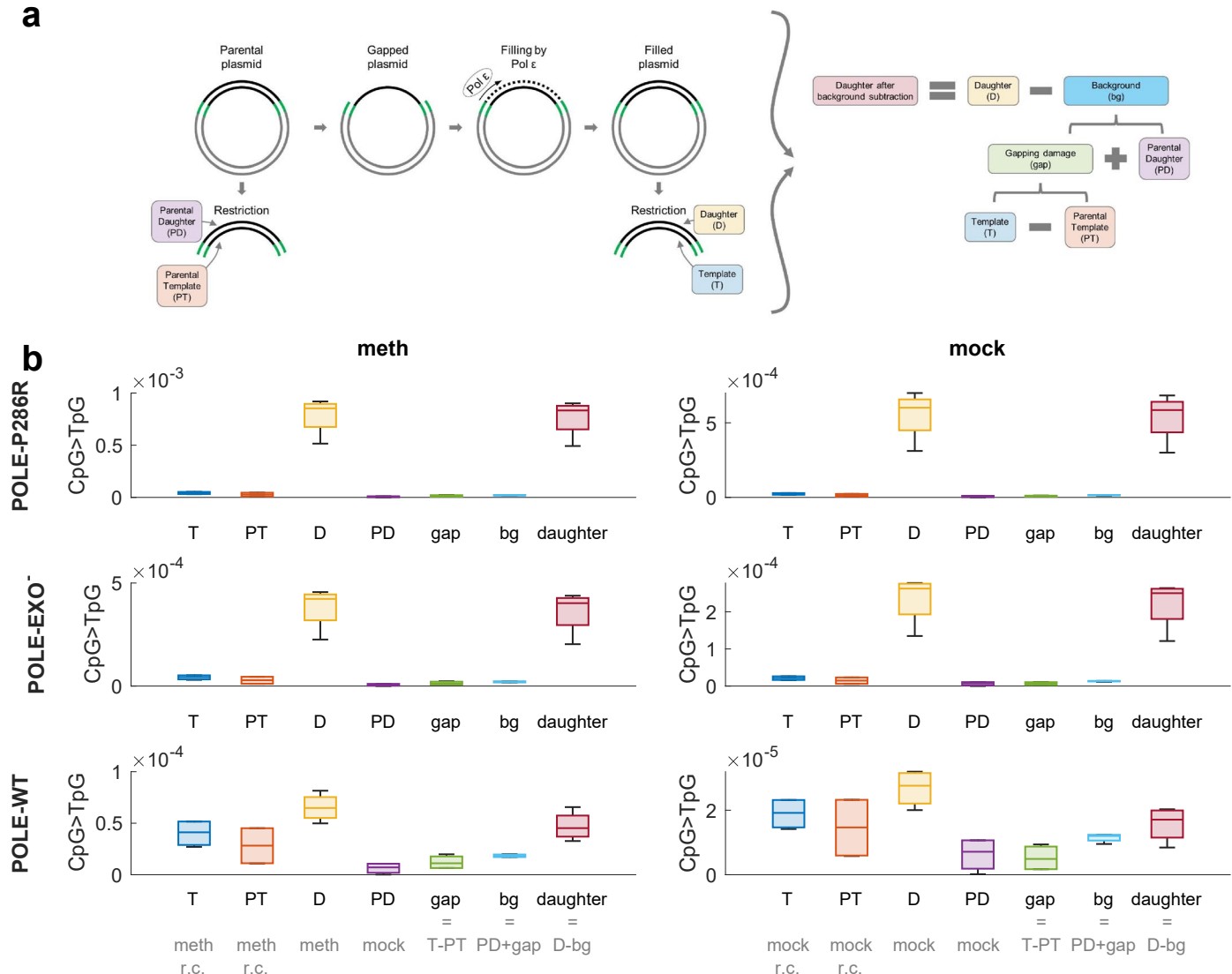

**Extended Data Fig. 7 | Background subtraction in PER-seq. a**, Diagram of the four strands sequenced in PER-seq (PD, PT, D, T) and how their values are used to determine the true-positive polymerase error rate in the daughter strand after background subtraction (dark red) by subtracting background (blue) from the raw mutation frequency in the daughter strand (yellow). The background then consists of two components: potential gapping damage (green) that could have happened to the template strand when single-stranded and before/while being filled, and a general background (purple) estimated by the raw mutation frequency in the parental daughter (PD) strand. Finally, the gapping damage is estimated as the difference between the template (T; darker blue) and parental template (PT; dark orange) strands. Of note, only fully filled molecules can undergo successful restriction digest and downstream library preparation for both the template and daughter strands, and therefore unfilled plasmids do not confound the results. In other words, by 'template' we mean the template strand of the ROI after filling by the respective polymerases. **b**, The CpG>TpG mutation frequency for all the values described in **a**. N = 4 replicates each. Boxplots are plotted with the MATLAB function boxchart (Methods).

# Reporting Summary

## Statistics

For all statistical analyses, confirm that the following items are present in the figure legend, table legend, main text, or Methods section.

| n/a | Confirmed | |
|---|---|---|
| ☐ | ☒ | The exact sample size (*n*) for each experimental group/condition, given as a discrete number and unit of measurement |
| ☐ | ☒ | A statement on whether measurements were taken from distinct samples or whether the same sample was measured repeatedly |
| ☐ | ☒ | The statistical test(s) used AND whether they are one- or two-sided<br>*Only common tests should be described solely by name; describe more complex techniques in the Methods section.* |
| ☐ | ☒ | A description of all covariates tested |
| ☐ | ☒ | A description of any assumptions or corrections, such as tests of normality and adjustment for multiple comparisons |
| ☐ | ☒ | A full description of the statistical parameters including central tendency (e.g. means) or other basic estimates (e.g. regression coefficient) AND variation (e.g. standard deviation) or associated estimates of uncertainty (e.g. confidence intervals) |
| ☐ | ☒ | For null hypothesis testing, the test statistic (e.g. *F*, *t*, *r*) with confidence intervals, effect sizes, degrees of freedom and *P* value noted<br>*Give P values as exact values whenever suitable.* |
| ☒ | ☐ | For Bayesian analysis, information on the choice of priors and Markov chain Monte Carlo settings |
| ☒ | ☐ | For hierarchical and complex designs, identification of the appropriate level for tests and full reporting of outcomes |
| ☒ | ☐ | Estimates of effect sizes (e.g. Cohen's *d*, Pearson's *r*), indicating how they were calculated |

*Our web collection on statistics for biologists contains articles on many of the points above.*

## Software and code

Policy information about availability of computer code

| Data collection | No software was used for data collection (the data sets were downloaded using standard tools, such as wget). |
|---|---|
| Data analysis | Most of the data analysis was carried out using MATLAB (2022a).<br><br>The following MATLAB toolboxes have been used:<br>- Statistics and Machine Learning Toolbox (version 12.3)<br><br>The following MATLAB library has been used:<br>- lbmap: Robert Bemis (2022). Light Bartlein Color Maps (https://www.mathworks.com/matlabcentral/fileexchange/17555-light-bartlein-color-maps), MATLAB Central File Exchange. Retrieved June 23, 2022.<br><br>The other software used:<br>- bedtools (version 2.27.0)<br>- fastqc (version 0.11.8)<br>- bowtie2 (version 2.3.5.1)<br>- samtools (version 1.9)<br>- GNU Wget 1.14<br>- picard (version 2.21.6)<br>- trimgalore (version 0.4.5)<br>- bismark (version 0.19.0) |

For manuscripts utilizing custom algorithms or software that are central to the research but not yet described in published literature, software must be made available to editors and reviewers. We strongly encourage code deposition in a community repository (e.g. GitHub). See our web collection on statistics for biologists.

The code developed in this study is available at https://bitbucket.org/licroxford/per-seq (PER-seq pipeline) and https://bitbucket.org/licroxford/cpg_mutagenesis (analysis and comparison of PER-seq and cancer data, all code to reconstruct the figures and tables in the study).

For manuscripts utilizing custom algorithms or software that are central to the research but not yet described in published literature, software must be made available to editors and reviewers. We strongly encourage code deposition in a community repository (e.g. GitHub). See the Nature Portfolio guidelines for submitting code & software for further information.

## Data

Policy information about availability of data

All manuscripts must include a data availability statement. This statement should provide the following information, where applicable:
- Accession codes, unique identifiers, or web links for publicly available datasets
- A description of any restrictions on data availability
- For clinical datasets or third party data, please ensure that the statement adheres to our policy

PER-seq sequencing data have been deposited in the Sequence Read Archive (SRA) under accession number SRP439101 and the processed files are available together with the code in the Bitbucket repositories. The used publicly available cancer samples are listed in Extended Data Table 5.

## Research involving human participants, their data, or biological material

Policy information about studies with human participants or human data. See also policy information about sex, gender (identity/presentation), and sexual orientation and race, ethnicity and racism.

| | |
|---|---|
| Reporting on sex and gender | NA |
| Reporting on race, ethnicity, or other socially relevant groupings | NA |
| Population characteristics | NA |
| Recruitment | NA |
| Ethics oversight | NA |

Note that full information on the approval of the study protocol must also be provided in the manuscript.

# Field-specific reporting

Please select the one below that is the best fit for your research. If you are not sure, read the appropriate sections before making your selection.

☒ Life sciences          ☐ Behavioural & social sciences          ☐ Ecological, evolutionary & environmental sciences

For a reference copy of the document with all sections, see nature.com/documents/nr-reporting-summary-flat.pdf

# Life sciences study design

All studies must disclose on these points even when the disclosure is negative.

| | |
|---|---|
| Sample size | Sample sizes indicated in individual figures, which represent typical values used in the field for this type of experiments. No sample size calculation was performed, however sample size is sufficient based on significance of relevant statistical tests applied. |
| Data exclusions | None of the experiments passing the technical experiment success criteria were excluded from the analyses |
| Replication | Experiments were replicated as indicated in the manuscript, appropriate statistical tests were used, and distributions and P values indicated. |
| Randomization | Randomization experiment design is not suitable for the presented experiments, because biochemical experiments do not employ sampling from the population. |
| Blinding | Blinding experiment design is not suitable for the presented experiments, because this study does not evaluate the effects of an exposure. |

# Reporting for specific materials, systems and methods

We require information from authors about some types of materials, experimental systems and methods used in many studies. Here, indicate whether each material, system or method listed is relevant to your study. If you are not sure if a list item applies to your research, read the appropriate section before selecting a response.

## Materials & experimental systems

| n/a | Involved in the study |
|---|---|
| ☐ | ☒ Antibodies |
| ☐ | ☒ Eukaryotic cell lines |
| ☒ | ☐ Palaeontology and archaeology |
| ☒ | ☐ Animals and other organisms |
| ☒ | ☐ Clinical data |
| ☒ | ☐ Dual use research of concern |
| ☒ | ☐ Plants |

## Methods

| n/a | Involved in the study |
|---|---|
| ☒ | ☐ ChIP-seq |
| ☒ | ☐ Flow cytometry |
| ☒ | ☐ MRI-based neuroimaging |

## Antibodies

| | |
|---|---|
| Antibodies used | POLE (Stratech; GTX132100-GTX); β-actin (Cell Signaling Technology; 3700); anti-mouse IgG (H + L)-HRP (Bio-Rad; 1706516); goat anti-rabbit IgG (H + L)-HRP (Bio-Rad; 1706515) |
| Validation | Multiple publications: Nature (PMID: 37968395), Cell Reports (PMID: 35649380), Cell (PMID: 35512704), J Med Gen (PMID: 35534205) |

## Eukaryotic cell lines

Policy information about cell lines and Sex and Gender in Research

| | |
|---|---|
| Cell line source(s) | E14 mESCs gift from Adrian Bird group, HCC2998 gift from Xin Lu group |
| Authentication | Non authenticated, WGS data provided for mESCs and could be used for verification of validity. |
| Mycoplasma contamination | Cells were regularly (monthly) tested for mycoplasma and found negative |
| Commonly misidentified lines (See ICLAC register) | None |

## Plants

| | |
|---|---|
| Seed stocks | *Report on the source of all seed stocks or other plant material used. If applicable, state the seed stock centre and catalogue number. If plant specimens were collected from the field, describe the collection location, date and sampling procedures.* |
| Novel plant genotypes | *Describe the methods by which all novel plant genotypes were produced. This includes those generated by transgenic approaches, gene editing, chemical/radiation-based mutagenesis and hybridization. For transgenic lines, describe the transformation method, the number of independent lines analyzed and the generation upon which experiments were performed. For gene-edited lines, describe the editor used, the endogenous sequence targeted for editing, the targeting guide RNA sequence (if applicable) and how the editor was applied.* |
| Authentication | *Describe any authentication procedures for each seed stock used or novel genotype generated. Describe any experiments used to assess the effect of a mutation and, where applicable, how potential secondary effects (e.g. second site T-DNA insertions, mosiacism, off-target gene editing) were examined.* |

