## [Peer Review File · Nature Genetics]

Peer Review Information

Manuscript Title: Human DNA polymerase ϵ is a source of C>T mutations at CpG dinucleotides

Corresponding author name(s): Professor Skirmantas Kriaucionis, Dr Benjamin Schuster-Böckler, Dr Marketa Tomkova

Editorial Notes:

Transferred manuscripts This document only contains reviewer comments, rebuttal and decision letters for versions considered at Nature Genetics.

Reviewer Comments & Decisions:

Decision Letter, initial version:

11th Aug 2023

Dear Professor Kriaucionis,

Your Article, "Human DNA polymerase ϵ fidelity is reduced when replicating methyl-CpGs" has now been seen by 4 referees. You will see from their comments copied below that while they find your work of considerable potential interest, they have raised quite substantial concerns that must be addressed. In light of these comments, we cannot accept the manuscript for publication, but would be very interested in considering a revised version that addresses these serious concerns.

Reviewer #1:

-validate the system and Pol ϵ mutants using *Xenopus* egg extracts or in cell lines

Reviewer #2:

-provide technical controls: enzyme specific activity/ DNA synthesis product quantification/ reaction conditions for Pol ϵ gap-filling reaction/ control for DNA synthesis on methylated DNA

Reviewer #3:

-discuss the longstanding controversy regarding Pol ϵ as the primary leading strand replicase in light of previous works on Pol δ

Reviewer #4:

- rationale for the choice of ROI length
- MMRd come before Poled?

We hope you will find the referees' comments and the prioritised set of referee points useful as you decide how to proceed. If you wish to submit a substantially revised manuscript, please bear in mind that we will be reluctant to approach the referees again in the absence of major revisions. Please do not hesitate to get in touch if you would like to discuss these issues further.

If you choose to revise your manuscript taking into account all reviewer and editor comments, please highlight all changes in the manuscript text file. At this stage we will need you to upload a copy of the manuscript in MS Word .docx or similar editable format.

- *1) Include a "Response to referees" document detailing, point-by-point, how you addressed each referee comment. If no action was taken to address a point, you must provide a compelling argument. This response will be sent back to the referees along with the revised manuscript.
- *2) If you have not done so already please begin to revise your manuscript so that it conforms to our Article format instructions, available here. Refer also to any guidelines provided in this letter.
- *3) Include a revised version of any required Reporting Summary: <https://www.nature.com/documents/nr-reporting-summary.pdf> It will be available to referees (and, potentially, statisticians) to aid in their evaluation if the manuscript goes back for peer review. A revised checklist is essential for re-review of the paper.

Please be aware of our guidelines on digital image standards.

[redacted]

If you wish to submit a suitably revised manuscript we would hope to receive it within 6 months. If

you cannot send it within this time, please let us know. We will be happy to consider your revision so long as nothing similar has been accepted for publication at Nature Genetics or published elsewhere. Should your manuscript be substantially delayed without notifying us in advance and your article is eventually published, the received date would be that of the revised, not the original, version.

Thank you for the opportunity to review your work.

Sincerely,
Chiara

Chiara Anania, PhD
Associate Editor
Nature Genetics
<https://orcid.org/0000-0003-1549-4157>

Referee expertise:

Referee #1: stem cell genomics

Referee #2: DNA polymerases; DNA replication; DNA damage and repair

Referee #3: DNA replication fidelity

Referee #4: Bioinformatics, Computational biology, RNA, Cancer genomics

Reviewers' Comments:

Reviewer #1:
Remarks to the Author:

In the current manuscript, Tomkova et al hypothesize that mutated Pol ϵ may represent the main mechanism for induction of C>T mutations at CpG dinucleotides (i.e., SBS1) in a deamination independent manner. This idea is based on their previous finding of increased CpG>TpG mutations in cancers expressing mutated Pol ϵ . The authors developed a new method to measure the error spectrum of polymerases. By using this PER-seq method, the authors show that mutated Pol ϵ (P286R) indeed results in an increased CpG>TpG mutation rate. Additionally, they find that wild-type

Pol ϵ has a 7-fold higher error rate when replicating methylated CpG islands compare to cytosines in other contexts. Together, Tomkova et al. claim that Pol ϵ dependent errors at CpG dinucleotides outnumber the commonly thought main mechanism underlying SBS1 mutations, namely spontaneous deamination. The main message of the manuscript is important as SBS1 is a very common and carcinogenic signature in cancers and healthy tissues. However, the authors do sometimes make big claims based on the data they present. Thus, some additional experiments and analyses are required before this manuscript can be published in Nature Genetics. The proposed major and minor revisions are detailed below.

Major points:

- Most claims in the current manuscript are based on PER-seq, a clean system where the only protein in the system is Pol ϵ , the protein of interest. In the manuscript, the authors directly compare these with in vivo WGS and WES data. Hereby the authors ignore all other proteins that may interact with the DNA and may change the mutational processes. One crucial experiment is shown in Figure 5a where the authors show that to get the same level of CpG>TpC mutations, which Pol ϵ introduces in one replication round, is generated by spontaneous deamination in 2.5 years. With this data the claim is made that it is therefore more likely that Pol ϵ errors underly SBS mutations instead of spontaneous deamination. Although the model is extremely interesting, as it explains the puzzling observations in cancer patient data as listed in the discussion, can this claim be made solely on the PER-seq data? 2.5 years is not that long in a human lifetime, and in our cells there are additional proteins that can repair mismatches, such as the base excision repair proteins TDG and MBD4. Indeed, germline loss of the latter protein causes a massive increase in CpG>TpC mutations in AML (Sanders et al., Blood 2018). One way to address this, is by validating the system and Pol ϵ mutants using Xenopus egg extracts, or in cell lines.
- Related to the latter point, in the leading vs lagging strand bias only the ratio is plotted. I would expect that spontaneous deamination is strand agnostic, whereas only the replication errors occur on the leading strand. One way to disentangle both processes is by plotting the contribution of CpG>TpC mutations on both strands. With the measurement on how many mutations accumulate by spontaneous deamination at 37 degrees C in 2.5 years, one could estimate if those numbers fit with what is observed in the contribution to both strands using the data depicted in Fig. 4c.
- The order of the manuscript is not intuitive. After Figure 1 it would make more sense to benchmark the system on WT Pol ϵ , compare this to the mutated variants and subsequently to online repositories (COSMIC). The data on WT Pol ϵ is now introduced in Figure 3, but it would be good to already show it in Figure 2 so that the background mutation burden of the system is known. Another example is that the make-up of the graphs depicted in Figures 1g&h is only explained in the text when Figures 3b-e are mentioned (page 6).
- In Figure 2, PER-seq is applied to mutant Pol ϵ . The signature refitting indeed shows contribution of SBS1, but also a large contribution of SBS5. This is also a clock-like signature but is not focused on in the manuscript. How can the authors explain such a large contribution of SBS5? Is it overfitting of the rather A-specific signature (for example the first 4 T>C peaks are not observed in the pattern depicted in Fig. 2a)?
- Can Pol δ also be tested in the PER-seq system? Do the authors expect a completely different mutational outcome that with Pol ϵ ? It would make their claim stronger that CpG>TpC mutations are

due to Pol ϵ activity.

- In the legend of Figure 4 it is stated that all analyses are performed on solely the exonic regions to increase the size of the dataset by merging WES and WGS data. It would be interesting to assess in the WGS data at all non-exonic regions to see if the pattern is the same there. This would exclude any transcription-based effects that may also cause mutations or increase the efficiency of spontaneous deamination because the DNA is single stranded at that time.

- How representative are the 2 ROI's for the meta-analyses of the human cancer data? Are these exonic regions? Is two ROI's sufficient to represent the entire human exome?

Minor points:

- The title of the manuscript suggests a study that is solely focused on WT Pol ϵ . However, the main message is that SBS1 is mainly caused by the replicative activity of Pol ϵ , which is shown by assessing mutant Pol ϵ . I would advise to change the title a bit so that it covers the main message.

- On page 5 when the median mutation rate is mentioned a unit is missing (342×10^{-6} per bp?).

- Figures 3c-e are plotted counterintuitively when considering Fig. 3b and reading the Discussion. Spontaneous deamination will lead to mC>T changes, whereas Pol ϵ errors will lead to a G>A change, because Pol ϵ will mis-incorporate against a mC residue. Yet in the Figures mainly CpG>TpG changes are seen (instead of GpC>ApC changes). Or is spontaneous deamination still a strong contributor?

- The analysis in Fig. 2d: can you do this? By only considering the 8 most contributing peaks, you are inflating the P-value, but what does that say? I would leave this out.

- Can a correlation be calculated for the graphs depicted in Fig. 4d? This will allow the author to calculate if there is a significant correlation between methylation status and the number of CpG>TpG changes.

Reviewer #2:

Remarks to the Author:

Tumors with mutations in POLE have some of the highest tumor mutation burdens measured with very unique mutation spectra. One of those unique mutations in CpG>TpG transition mutations, the mechanism of which is unresolved. One possibility is that the mutant POLE itself has a reduced replication fidelity on methylated cytosines. The authors use a high throughput sequencing method in vitro with three different human Pol epsilon constructs (wild type, exo-deficient, and P286R) to measure DNA replication fidelity on methylated DNA. They show that the P286R mutant causes significantly higher CpG>TpG mutagenesis than exo- or wt Pol epsilon and that the spectrum of errors resembles that seen in human tumors. They further show that the wild type Pol epsilon itself has a higher intrinsic error rate on methylated cytosines.

Conclusive demonstration that Pol epsilon and replication errors is a significant driver of CpG>TpG mutagenesis in both POLE mutant and POLE wt tumors would be a novel and highly significant finding. The technique used here is very powerful and also potentially very useful. However, there are some major issues with the interpretation of the results that would need to be addressed.

A major complicating factor in interpreting the results is the absence of specific activity measurements for each enzyme prep and, related, lack of quantification of fully extended DNA synthesis product. The current assay design only sequences Sac/Hind-sensitive fragments, or those that have had the gap fully synthesized. Partially filled substrates are excluded. Extended Fig 2a does show qualitative differences in digestion, and this is also semi-quantitative. The P286R lanes appear fully digested while *exo-* and *wt* have residual resistant DNA. The observed increase could thus represent the intrinsic activity of the enzyme. Shcherbakova has shown that the yeast P301R (P286R equivalent) mutation makes a hyperactive DNA polymerase (Xing et al Nat Comms 2019). This would be consistent with the observed P286R>*exo-*>*wt*.

The mechanistic explanation described requires two different polymerase misinsertion events across cytosine: dT opposite C and dA opposite mC (as nicely shown in Fig 3b). A simpler explanation is that a single misinsertion event can explain the *in vitro* observations. The observed NCT>NAT is due to dT opposite C and represents the true POLE mutation signature (NCT>NAT). While the NCG>NTG could be due to mCG deaminated to TG followed by correct dA insertion opposite the now template T. The authors do correct for possible deaminated cytosines in their template (Ext Fig 7), but the differences in enzyme prep activity complicate this interpretation. The authors go to great lengths to quantitate the deamination of mC in dsDNA, but it is well known that ssDNA spontaneously deaminates at significantly faster rates than in dsDNA, and the substrate used in these experiments spends time as ssDNA. 100% of the template molecules are interrogated in the mock-treated whereas some smaller fraction of extended molecules is interrogated in the Pol epsilon-“treated” molecules. If this truly is a replication fidelity difference, that the mutant DNA polymerase misinserts different nucleotides based on the 5' functional group of cytosine, this really needs to be shown kinetically using the tools that this group has assembled.

Another complicating factor is that Shcherbakova et al have also shown that the P301R enzyme is actually more faithful on a naked DNA template than the *exo-* POLE (Fig 1C, Xing et al Nat Comms 2019). This is inconsistent with the results shown here and not discussed.

The authors claim that 100% of CpG sites are methylated. It follows then that all CpG sites on a single template molecule should also be methylated. This reviewer calculates that the p53 target site should then have eight NpCpG sites with methylated cytosines. It would be incredibly informative to see this fidelity resolved at the sequencing level. For example, what fraction of reads have 0, 1, 2, 3, etc (or even all 8) CpG>TpG changes in a single read?

The reaction conditions for the Pol epsilon gap-filling reactions are not provided. The fidelity of all DNA polymerases is sensitive to salt, dNTP, Mg²⁺, pH conditions.

As exciting as this technique potentially is, there is no control DNA synthesis on methylated DNA. It would be incredibly helpful to show that the CpG mutation spectrum is not seen with an unrelated pol, like Klenow-*exo-* for example.

Minor points

I believe in Fig 1D the authors mean to cite Jee et al 2016.

The probability density function is not well explained.

Reviewer #3:

Remarks to the Author:

Given that C to T transitions in CpG dinucleotides are among the most common mutations found in human cancers and genetic diseases, the key results of this study are:

1. Pol ϵ is a major source of CpG>TpG errors.
2. Wild type Pol ϵ has a significantly higher error rate when replicating 5-methylcytosine (5mC)-containing CpG sites compared to replicating C in other contexts.
3. The researchers developed a new method called Polymerase Error Rate Sequencing (PER-seq) to measure the error spectrum of DNA polymerases in vitro. (This is very exciting on a personal level.)
4. The most common human cancer-associated Pol ϵ mutant (P286R) is found to produce an excess of CpG>TpG errors, mimicking the mutation pattern seen in tumors with this mutation and deficiencies in mismatch repair.
5. In human cancer samples, regions with higher MMR activity show fewer CpG>TpG mutations, suggesting that replication errors are a major contributor to CpG>TpG mutagenesis.
6. In most healthy tissues, Pol ϵ -derived CpG>TpG mutations likely outnumber those due to 5mC deamination.

I find the new technique to be novel and well presented. Many of the conclusions are quite novel and even surprising (see point #6 above). The data are well presented: the figures are clear; the logic is laid out well; and there is little to no overreach in the conclusions. The Methodologies are clearly stated and appear to be appropriate. The statistics are thorough and also appropriate. The Conclusions are robust and follow naturally from the data.

I would like to see some changes to Fig 1c:

I suspect that the data exists and no further experiments will be required. If the probability of the same artefact at the same position is $\sim 10^{-9}$, then a version of Fig 1c with dilutions out to 10^{-9} or 10^{-10} would be illuminating. It's important to show where a new assay breaks down as well as where it shines.

I would suggest an addition to the Discussion section:

(Note from reviewer: papers cited herein are for the benefit of the authors; many are already cited in this manuscript or in reviews cited by this manuscript; manuscript acceptance is not contingent on adding such citations to the next draft.)

The authors clearly know that mutation biases were used to assign Pol epsilon the role of primary leading strand replicase. However, this assignment has been called into question and the authors have a golden opportunity to explicitly weigh in. The biases for complementary mispairs in Fig 3c-f closely match those found in yeast with either engineered mutator pol2-M644G (Fig 2; PMID: 25217194) or exo- pol2-04 (Supp. Fig. 2, PMID: 34887558). Such mutation biases, alongside ribonucleotide incorporation maps (PMIDs: 25622106, 25622295, 25624100, 26492137, 31488849, 36434012), were used to assign the leading/lagging responsibilities of the replicases across cerevisiae, pombe, and human nuclear genomes. Most of these biases were seen in Kunkel's earlier M13 LacZ assays. However, those assays had some artefacts due to, presumably, oxidative damage. Some believe Pol epsilon incapable of replicating the bulk of the leading strand (PMID: 26186286) and have used misinterpretations of said artefacts to call into question the comparability of in vitro/in vivo mutation spectra (PMID: 26145172). They claim that Pol epsilon is primarily an extrinsic proofreader of Pol delta errors and thus that mutations in POLED & MMRd cancers are made by Pol delta (PMID: 37307920). Your data lacks oxidative damage artefacts and shows that the mutation signature in POLED & MMRd cancer is fully explainable by Pol epsilon errors. This includes the T-dT mispairs that were key to the first leading strand assignment of Pol epsilon (PMID: 17615360) but were recently

claimed NOT to be due to Pole epsilon (PMID: 37307920 again). You may be uniquely positioned to end the controversy once and for all. For the good for the field (and a potential boost to readership and impact), please add a brief discussion of the controversy and how your data puts the final nail in the coffin.

Reviewer #4:

Remarks to the Author:

Mutations are essential for evolution. Understanding how these mutations arise would provide fundamental insights into the biological processes active in the living system. In this manuscript, the authors developed a novel method (PER-seq) to understand the type and frequency of DNA mismatches introduced by DNA Pol epsilon (both wild-type and mutant), one of the major human replicative DNA polymerases, at different DNA bases and the influence of neighbouring sequence context. They observed that the mismatches, which eventually lead to C>T mutations, are highly enriched at CpG sites, especially when the C is methylated. The results obtained from this PER-seq are corroborated with the somatic mutations spectrum observed in cancer patients with mutant DNA Pol epsilon and deficient MMR. In addition, they reported an interesting finding that the fidelity of wild-type Pol epsilon at non-methylated CpG is low, in general, suggesting that the C>T mutations at these sites could occur beyond the epigenomics modification found on CpG. Overall, the manuscript is well written and the findings support the conclusion. The following are a few minor comments/suggestions for improvement:

Major comments:

- 1) Figure 1a: What is the rationale for the choice of ROI length, ~200bp? Is this size sufficient to capture the DNA polymerase activity well (given the speed of replication)?
- 2) Figure 1d, the range of mutation frequency (from Lee et al., 2016) is not overlapping for all mutation types. For example, the range of C>T and G>T are quite high in the previous study than the PER-seq.
- 3) Figure 2a-b: although it is interesting to see that the mutation profile of the PER-POLE-P286R closely resembles (with high Cosine similarity) with the mutation profiles of POLEd & MMRd tumours. However, in tumours, the mutation profile was probably shaped by deficiency of both POLE and MMR at different time points during the clonal evolution of the cancers. In other words, the deficiency of MMR occurred before POLE, or vice versa, but based on the similarity with the PER-POLE-P286R does it suggest that the tumours shown here harboured MMRd first and then POLEd?
- 4) For the above analysis, it would also be interesting to see mutation profiles in tumours with germline biallelic mismatch repair deficiency (bMMRd) followed by somatic POLE deficient events (eg., <https://doi.org/10.1038/ng.3202>).

Minor comment:

- 1) In Figure 1c, please specify what is being plotted on the x- and y-axis labels, is it observed (or expected) mismatched or mutations?
- 2) In the Methods section, under DNA polymerases: "hon-human" -> "non-human"

Author Rebuttal to Initial comments
--

Reviewer #1:

Remarks to the Author:

In the current manuscript, Tomkova et al hypothesize that mutated Pol ϵ may represent the main mechanism for induction of C>T mutations at CpG dinucleotides (i.e., SBS1) in a deamination independent manner. This idea is based on their previous finding of increased CpG>TpG mutations in cancers expressing mutated Pol ϵ . The authors developed a new method to measure the error spectrum of polymerases. By using this PER-seq method, the authors show that mutated Pol ϵ (P286R) indeed results in an increased CpG>TpG mutation rate. Additionally, they find that wild-type Pol ϵ has a 7-fold higher error rate when replicating methylated CpG islands compare to cytosines in other contexts. Together, Tomkova et al. claim that Pol ϵ dependent errors at CpG dinucleotides outnumber the commonly thought main mechanism underlying SBS1 mutations, namely spontaneous deamination. The main message of the manuscript is important as SBS1 is a very common and carcinogenic signature in cancers and healthy tissues. However, the authors do sometimes make big claims based on the data they present. Thus, some additional experiments and analyses are required before this manuscript can be published in Nature Genetics. The proposed major and minor revisions are detailed below.

Major points:

- Most claims in the current manuscript are based on PER-seq, a clean system where the only protein in the system is Pol ϵ , the protein of interest. In the manuscript, the authors directly compare these with in vivo WGS and WES data. Hereby the authors ignore all other proteins that may interact with the DNA and may change the mutational processes. One crucial experiment is shown in Figure 5a where the authors show that to get the same level of CpG>TpC mutations, which Pol ϵ introduces in one replication round, is generated by spontaneous deamination in 2.5 years. With this data the claim is made that it is therefore more likely that Pol ϵ errors underly SBS mutations instead of spontaneous deamination. Although the model is extremely interesting, as it explains the puzzling observations in cancer patient data as listed in the discussion, can this claim be made solely on the PER-seq data? 2.5 years is not that long in a human lifetime, and in our cells there are additional proteins that can repair mismatches, such as the base excision repair proteins TDG and MBD4. Indeed, germline loss of the latter protein causes a massive increase in CpG>TpC mutations in AML (Sanders et al., Blood 2018). One way to address this, is by validating the system and Pol ϵ mutants using *Xenopus* egg extracts, or in cell lines.

We completely agree with the reviewer's comments about the importance of the different repair pathways and other proteins that influence the mutational process. **We have changed the explanation in the legend for Fig 5 and in the Results to emphasize that our comparison of deamination rate and Pol ϵ error rate refers to the source of the mutations, i.e., mismatches (referred to as errors) which can become mutations because of DNA repair or replication.** We also now use a clearer notation for mis-incorporation events (C:dA) as opposed to mutations (C>T). We have also changed the title to "Human DNA polymerase ϵ is a source of C>T mutations at CpG dinucleotides". The data in Figure 5a are intended to estimate how Pol ϵ -derived errors compare to deamination-derived errors. As we state in the discussion, this does not rule out differences in the efficiency of repair of those two distinct types of errors.

The reviewer made the great suggestion to measure Pol ϵ errors in a more complex and natural environment, such as cell-extracts. Following this recommendation, we have now added several

additional experiments and analyses to further support that our PER-seq results capture a real biological phenomenon:

1) Additional *in vitro* evidence that Pol ϵ is a source of CpG>TpG mutations

First, we engineered the P286R mutation into the endogenous Pol ϵ gene in a mouse embryonic stem cell line using CRISPR/Cas9, passaged cells for two months to enable accumulation of mutations, and performed whole genome sequencing on single-cell-derived clones to determine the resulting mutational pattern and rate of CpG>TpG mutations compared to Pol ϵ proficient cells (new Fig. 5a-d). We observed a strong similarity between the cell-line derived mutational signature and our PER-seq P286R signature, including a high rate of CpG>TpG mutations in the Pol ϵ P286R cells (but not in Pol ϵ proficient cells), as well as elevated mutations in methylated CpGs. Crucially, we observe clear leading strand bias (new Fig. 5d), in line with these mutations resulting from 5mC:dA errors by Pol ϵ .

2) Additional *in vivo* evidence that Pol ϵ is a source of CpG>TpG mutations

Furthermore, we have added a new section of the study to validate that Pol ϵ is a source of CpG>TpG mutations *in vivo*. To do so, we made use of whole-exome sequencing data from a published mouse model carrying P286R and S459F alleles (<https://doi.org/10.1158/0008-5472.can-20-0624>). We show that the rate of CpG>TpG mutations in these mice is high and exhibits clear *leading strand* bias and enrichment at methylated CpGs (new Fig. 5e-g).

3) Evidence that replication errors at 5mC also occur in a cellular context using nuclear extracts

We developed PER-EXTRACT-Seq to measure errors of polymerases in nuclear extracts of the same Pol ϵ mutant and *wt* cell lines. The measured profiles are expected to be less clear, as multiple polymerases may contribute to them, and they will be shaped by the repair machinery and potentially other sources of damage. Nevertheless, our results show an elevated 5mC:dA error rate in extracts from Pol ϵ mutant cells, in line with the higher CpG>TpG mutation rate in the corresponding cells. This experiment demonstrates that endogenously produced enzymes in the presence of accessory proteins introduce errors when replicating methylated CpGs (new Supplementary Figure 6).

These additional experiments have now been integrated into the manuscript under a new sub-heading "POLE-P286R is sufficient to cause a high CpG>TpG mutation rate in cells and *in vivo*".

- Related to the latter point, in the leading vs lagging strand bias only the ratio is plotted. I would expect that spontaneous deamination is strand agnostic, whereas only the replication errors occur on the leading strand. One way to disentangle both processes is by plotting the contribution of CpG>TpC mutations on both strands. With the measurement on how many mutations accumulate by spontaneous deamination at 37 degrees C in 2.5 years, one could estimate if those numbers fit with what is observed in the contribution to both strands using the data depicted in Fig. 4c.

As we clarified in our reply above, the deamination rate at 37°C merely represents the frequency at which T:G mismatches will appear in DNA. As the reviewer rightly pointed out, there are numerous independent repair pathways that will affect how many of these errors become fixated into a mutation. Unfortunately, we do not know the region-dependent rate and accuracy of repair of T:G mismatches. Modelling the mutation rate on the lagging strand by the estimated deamination rate would require precisely such knowledge of the quantitative impact of different repair mechanisms, as well as higher resolution and more tissue-specific maps of the leading/lagging strand direction for each of the respective cancer types, which are - to our knowledge - not available (limitations of the current gold-standard replication strand maps are described in <https://doi.org/10.1016/j.cell.2015.12.050>).

- The order of the manuscript is not intuitive. After Figure 1 it would make more sense to benchmark the system on WT Pol ϵ , compare this to the mutated variants and subsequently to online repositories (COSMIC). The data on WT Pol ϵ is now introduced in Figure 3, but it would be good to already show it in Figure 2 so that the background mutation burden of the system is known. Another example is that the make-up of the graphs depicted in Figures 1g&h is only explained in the text when Figures 3b-e are mentioned (page 6).

We edited the manuscript to explain the graphs depicted in Fig. 1g&h earlier.

The overall order in which to present the results is indeed a topic that we have discussed at length during the preparation of the manuscript. There are different ways in which the manuscript could be structured, but we ultimately converged on the current order, for the following reasons:

1. First, we introduce the PER-seq method. Here, we (a) show the background mutation burden of the system (which is given by the measurements in parental plasmids, we have now added a direct reference to the figure SFig. 2), (b) validate the system with artificially mutated plasmids, and (c) benchmark the assay against a low-fidelity polymerase for which base-specific error rates have previously been reported (Klenow-exo-) and another polymerase which is high-fidelity (KapaU+). This demonstrates the ability of PER-seq to accurately measure polymerase error rates.
2. We decided to follow this with the investigation of POLE-P286R because it has the highest error rate and a well-characterised mutational signature in the human cancer data. Presenting data from mutant polymerase first achieves two aims: (a) it shows that the PER-seq derived error profile matches mutation patterns observed in cancer patients, thus further supporting the validity of our measurements and (b) it also demonstrates how different SBS signatures can be attributed to the isolated activity of a polymerase.
3. Now that we introduced the method and how it applies to observations from cancer, we focus on the research question whether the high rate of CpG>TpG mutations in the POLEd & MMRd patients is due to Pol ϵ errors, as it logically follows from the previous point.
4. We focus on the WT Pol ϵ only afterwards, to address a different – but also very interesting question – whether perhaps Pol ϵ errors could contribute to CpG>TpG mutations even in POLE-proofreading proficient cells.
5. Finally, we focus on the role of DNA modifications and general implications of our discovered new source of CpG>TpG mutations.

While we understand that other arrangements of the results are possible, we think the current order is the best of all we tried. Notably, we received positive feedback on the presentation and clarity from other reviewers (esp. reviewer 3 and 4), thus we propose to keep the overall structure as it is. However, we have now changed the text to introduce WT Pol ϵ sooner and better explain the motivation for focusing on P286R first:

We performed PER-seq on methylated ROIs with *wt* human Pol ϵ (PER-POLE-WT), Pol ϵ containing the P286R mutation in the proofreading domain (PER-POLE-P286R) and Pol ϵ with a catalytically inactive (D275A; E277A) exonuclease (proofreading) domain (PER-POLE-EXO). We initially focused on POLE-P286R since it is the most common pathogenic *POLE* mutation observed in human cancers and mutational patterns resulting from mutated enzyme have been analysed before²⁸.

- In Figure 2, PER-seq is applied to mutant Pol ϵ . The signature refitting indeed shows contribution of SBS1, but also a large contribution of SBS5. This is also a clock-like signature but is not focused on in

the manuscript. How can de authors explain such a large contribution of SBS5? Is it overfitting of the rather a-specific signature (for example the first 4 T>C peaks are not observed in the pattern depicted in Fig. 2a)?

This is a very interesting question. We had a discussion paragraph about SBS5 included in our previous draft, but removed it due to length constraints. We have now again included it in the Discussion:

The cause of SBS5 is currently unexplained. Our data raise the possibility that polymerase errors are involved in the aetiology of SBS5, which would agree with its clock-like properties [REF CLOCK-LIKE]. In line with this possibility, the highest burden of SBS5 can be observed in POLEd and MMRd cancer patients, however, future research will be needed to determine whether polymerase errors might underlie SBS5 (Supplementary Note 10).

And we have added further details and discussion as the Supplementary Note 10:

The decomposition of the PER-POLE-P286R error signature into COSMIC SBS signatures showed a substantial contribution by SBS5 (Fig. 2d). The cause of SBS5 is currently unexplained. Our data raise the possibility that polymerase errors are involved in the aetiology of SBS5. This would agree with the clock-like properties of SBS5^{18,19}. We have explored this possibility and compared the SBS5 exposure in different cancer samples using SBS exposures reported in the PWCAG study¹⁹, downloaded from the PCAWG ICGC data portal. Interestingly, the highest burden of SBS5 can be indeed observed in POLEd and in MMRd cancer patients, showing more than 17-fold increase over exposures in patients proficient in POLE and MMR (Supplementary Figure 9). This supports a possibility that polymerase errors contribute to SBS5.

However, SBS5 has a very “flat” profile, and is therefore difficult to distinguish from other signatures with a relatively uniform mutation rate across sequence contexts⁷⁰. It is therefore possible that the “flat” component in PER-POLE-P286R is independent of SBS5. Finally, two of the distinguishing T>C peaks in SBS5 overlap with the two trinucleotides (TAT and AAT) not covered in our currently used two ROIs (Supplementary Figure 10). Therefore, future research will be needed to determine whether polymerase errors might contribute to SBS5.

Supplementary Figure 9: Exposure to SBS5 in POLEd or MMRd cancer patients in comparison with PROF (POLEp & MMRp) patients. The SBS5 exposures have been downloaded from the PCAWG ICGC data portal and matched to the samples used in this study. For a fair comparison, only cancer types with POLEd or MMRd samples are shown here in both groups.

- Can Pol δ also be tested in the PER-seq system? Do the authors expect a completely different mutational outcome than with Pol ϵ ? It would make their claim stronger that CpG>TpC mutations are due to Pol ϵ activity.

While it is not unreasonable to expect similarities between Pol ϵ and Pol δ , there is very little firm data to base any speculation on. Pol δ could be measured with PER-seq, and we plan to attempt this in the future. However, this is a project that will likely take several years, not least because purification of functional Pol δ holoenzyme is not trivial. It is therefore unfortunately out of scope for the current study.

- In the legend of Figure 4 it is stated that all analyses are performed on solely the exonic regions to increase the size of the dataset by merging WES and WGS data. It would be interesting to assess in the WGS data at all non-exonic regions to see if the pattern is the same there. This would exclude any transcription-based effects that may also cause mutations or increase the efficiency of spontaneous deamination because the DNA is single stranded at that time.

We have now included new Extended Data Figures 3-4 that shows the same analysis as in Fig. 4, but on WGS samples in the entire genome and in non-exonic regions, respectively. The overall conclusions remain the same (high frequency of CpG>TpG in POLED & MMRd, enrichment on the leading strand, and correlation with methylation levels), confirming that Fig. 4 conclusions were not driven by transcription-based effects.

- How representative are the 2 ROI's for the meta-analyses of the human cancer data? Are these exonic regions? Is two ROI's sufficient to represent the entire human exome?

Both ROIs are exonic regions and they were chosen to cover a wide spectrum of trinucleotides, with most being represented multiple times, as shown in the new Supplementary Figure 10. We chose natural sequences that exist in the human genome to avoid potential artefacts which could emerge in artificially designed sequence. The DNMT1 exonic CpG island was chosen to capture a CpG-rich region, to enable sufficient statistical power for the CpG-related questions. The choice of the TP53 ROI was based on it being one of the most important and highly mutated regions in cancer. We have now better explained the rationale for the ROI choice in the Methods and added a Supplementary Note 8, where we show that loci with high error rates in the PER-seq experiment match known TP53 hotspot mutations in human cancer.

Minor points:

- The title of the manuscript suggests a study that is solely focused on WT Pol ϵ . However, the main message is that SBS1 is mainly caused by the replicative activity of Pol ϵ , which is shown by assessing mutant Pol ϵ . I would advise to change the title a bit so that it covers the main message.

We thank for this feedback. On reflection, we decided to change the title to

“Human DNA polymerase ϵ is a source of C>T mutations at CpG dinucleotides”

- On page 5 when the median mutation rate is mentioned a unit is missing (342×10^{-6} per bp?).

We have added the unit (per bp).

- Figures 3c-e are plotted counterintuitively when considering Fig. 3b and reading the Discussion. Spontaneous deamination will lead to mC>T changes, whereas Pol ϵ errors will lead to a G>A change, because Pol ϵ will mis-incorporate against a mC residue. Yet in the Figures mainly CpG>TpG changes are seen (instead of GpC>ApC changes). Or is spontaneous deamination still a strong contributor?

In all the figures and text, we refer to the template base (and its sequence context). In order to avoid confusion, we have now added a new notation into the figures, for example C:dA, to clarify that these represent misincorporation of adenine opposite template cytosine.

- The analysis in Fig. 2d: can you do this? By only considering the 8 most contributing peaks, you are inflating the P-value, but what does that say? I would leave this out.

This was to show that the similarity is driven by the major peaks in both signatures (as cosine similarity can be strongly affected by background values in the two vectors). We have left Fig. 2d out now to avoid confusion.

- Can a correlation be calculated for the graphs depicted in Fig. 4d? This will allow the author to calculate if there is a significant correlation between methylation status and the number of CpG>TpG changes.

We have now added p-values (between low-mod vs high-mod CpGs) into Fig. 4d and Spearman correlation values as **Extended Data Figure 2e**. Both types of statistical comparison confirm a very significant relationship between methylation status and the frequency of CpG>TpG mutations.

Reviewer #2:

Remarks to the Author:

Tumors with mutations in POLE have some of the highest tumor mutation burdens measured with very unique mutation spectra. One of those unique mutations in CpG>TpG transition mutations, the mechanism of which is unresolved. One possibility is that the mutant POLE itself has a reduced replication fidelity on methylated cytosines. The authors use a high throughput sequencing method in vitro with three different human Pol epsilon constructs (wild type, exo-deficient, and P286R) to measure DNA replication fidelity on methylated DNA. They show that the P286R mutant causes significantly higher CpG>TpG mutagenesis than exo- or wt Pol epsilon and that the spectrum of errors resembles that seen in human tumors. They further show that the wild type Pol epsilon itself has a higher intrinsic error rate on methylated cytosines.

Conclusive demonstration that Pol epsilon and replication errors is a significant driver of CpG>TpG mutagenesis in both POLE mutant and POLE wt tumors would be a novel and highly significant finding. The technique used here is very powerful and also potentially very useful. However, there are some major issues with the interpretation of the results that would need to be addressed.

A major complicating factor in interpreting the results is the absence of specific activity measurements for each enzyme prep and, related, lack of quantification of fully extended DNA synthesis product. The current assay design only sequences Sac/Hind-sensitive fragments, or those that have had the gap fully synthesized. Partially filled substrates are excluded. Extended Fig 2a does show qualitative differences in digestion, and this is also semi-quantitative. The P286R lanes appear fully digested while exo- and wt have residual resistant DNA. The observed increase could thus represent the intrinsic activity of the enzyme. Shcherbakova has shown that the yeast P301R (P286R equivalent) mutation makes a hyperactive DNA polymerase (Xing et al Nat Comms 2019). This would be consistent with the observed P286R>exo->wt.

Specific activity of the protein was measured routinely, and any protein purification batches with substantially lower template-filling capacity were discharged. It is important to note that the amount of enzyme was titrated to achieve close-to-complete synthesis of the product. We have now added an entire section Supplementary Note 5 with specific activity measurements for each enzyme prep and quantification of fully extended DNA synthesis product (see below).

Importantly, while the wt, exo-, and P286R enzymes might differ in processivity, this does not confound the results: as only fully filled plasmids are used for measurements in the template and daughter strands, the interpretation of the error profiles of the different polymerases is independent of any variation in processivity.

Supplementary Note 5

The tables below represent percentage of unfilled plasmids:

Filling 1	Wt	exo-	P286R
TP53 unmethylated	ND	<10%*	ND
TP53 methylated	11.1%	<10%*	ND
DNMT1 unmethylated	0.9%	7.8%	ND
DNMT1 methylated	9.8%	12.2%	ND

Filling 2	wt	exo-	P286R
TP53 unmethylated	6%	12.6%	8.6%
TP53 methylated	<10%*	2%	<10%*

Filling 3	wt	exo-	P286R
TP53 unmethylated	ND	4.6%	ND
TP53 methylated	0.2%	3.1%	ND

Notes:

- The numbers represent the percentage of the template remaining unextended, calculated by scanning the gels and then using Image J on the raw TIFFs.
- ND = none detected (above background).
- * Exact data not available.

The table below shows specific activity measurements for each enzyme prep (normalised to wt):

	wt	exo-	P286R
First prep	1	1.01	0.88
Second prep	1	1.57	1.49

Notes:

- Specific activity was measured by performing extension reactions as described previously¹. Briefly, 20 fM of DNA polymerase was combined with excess of A2 substrate (CGCTGGCCGTAGTCTTCCAACGTCGTGACTGGGAAAA) annealed to C700/800 primer (TTTTCCAGTCACGACGTTG) and incubated over a time course (1-8 min). Extension products were resolved using denaturing electrophoresis and quantified using Licor Odyssey CLx imaging system. Values were normalised to activity of wt protein.

The table below shows mapping between enzyme prep, filling number, and NovaSeq libraries:

library	ROI	Enzyme prep	Filling
novaseq1	TP53	1	1
novaseq1	DNMT1	1	1
novaseq3	TP53	2	2
novaseq4	TP53	2	3

The mechanistic explanation described requires two different polymerase misinsertion events across cytosine: dT opposite C and dA opposite mC (as nicely shown in Fig 3b). A simpler explanation is that a single misinsertion event can explain the in vitro observations. The observed NCT>NAT is due to dT opposite C and represents the true POLE mutation signature (NCT>NAT). While the NCG>NTG could be due to mCG deaminated to TG followed by correct dA insertion opposite the now template T. The authors do correct for possible deaminated cytosines in their template (Ext Fig 7), but the differences in enzyme prep activity complicate this interpretation. The authors go to great lengths to quantitate the deamination of mC in dsDNA, but it is well known that ssDNA spontaneously deaminates at significantly faster rates than in dsDNA, and the substrate used in these experiments spends time as ssDNA. 100% of the template molecules are interrogated in the mock-treated whereas some smaller

fraction of extended molecules is interrogated in the Pol epsilon-“treated” molecules. If this truly is a replication fidelity difference, that the mutant DNA polymerase mis-inserts different nucleotides based on the 5’ functional group of cytosine, this really needs to be shown kinetically using the tools that this group has assembled.

We thank the reviewer for this comment which, we believe, highlights that we did not adequately explain the PER-seq protocol in the previous draft. The reviewer is absolutely correct that ssDNA will accumulate deamination events. We designed the PER-seq protocol with this in mind: we specifically sequence the template strand of the ROI *after* filling by the respective polymerases. This is what we refer to as the “template”. Any deamination pattern that was introduced during the gapping, filling, or library preparation will be detected therein. We do indeed see some CpG>TpG mutations on the template strand due to deamination of the ssDNA, and their frequency corresponds to the duration the substrate spends as ssDNA before filling (see below, panel c, which shows that the measured data in boxplots match the expected values shown as lines).

Additionally, we also sequence what we call the “parental” sample, that is the plasmid *before* the gapping & filling (see below panels a-b). We optimised the protocol to minimise heating-associated cytosine deamination during library preparation, which resulted in a substantial reduction of C>T deamination events emerging from construction of sequencing libraries (compare a and b below, see Supplementary Note 3).

This demonstrates that we can accurately detect deamination damage that happened both before and during library preparation.

Figure legend: Background C>T errors in parental samples from the old protocol (validation1 and novaseq1 libraries), new protocol (novaseq 3-5), and on the gapped template. The lines represent expected numbers based on the deamination rate of ssDNA from Ehrlich et al, 1986, <https://doi.org/10.1007/BF0116426>, and durations that ssDNA spends at different temperatures.

Crucially, ssDNA-induced mutations on the template are then subtracted from the pattern observed on the daughter strand, to ensure they do not contribute to the POLE error spectra. It is also worth pointing out that these background errors (e.g., cytosine deamination in gapped template) remain similar for POLE-WT, POLE-EXO⁻, and POLE-P286R:

Figure legend: Background C>T mutations in the template samples (corresponding to cytosine deamination during gapping, filling, and library preparation). For clarity, the raw mutation frequencies (without background subtraction) in the TP53 ROI are shown here.

This is in stark contrast to the true C:dA misincorporations detected in the daughter strand (i.e., misincorporation of dA opposite C by Pol ϵ), which show a ca. 6-fold increase in POLE-EXO⁻ and over 12-fold increase in POLE-P286R, compared to POLE-WT:

Figure legend: Mutations in the daughter samples (C:dA misincorporation). For clarity, the raw mutation frequencies (without background subtraction) in the TP53 ROI are shown here.

This shows clearly that the deamination rate of the template strand cannot explain the observed high CpG>TpG (C:dA) rate in the daughter strands and that these represent true mis-incorporations of adenine opposite template 5mC by the polymerase.

Finally, we would like to clarify that both the daughter and the template strands get sequenced only from those molecules that were completely filled, ensuring that enzyme prep activity does not confound this. Also as discussed above, we have measured specific activity of different enzyme batches to ensure that our enzyme purification is of consistent quality.

We have now included a new section Supplementary Note 4 with explanation above, and made changes to the manuscript and supplementary materials to clarify how PER-seq works.

Changes to the main text:

One strand of the ROI is then selectively removed, and the resulting single-stranded region is filled by a polymerase of interest (based on^{35,36}, see Methods for more details) (Fig. 1a, Extended Data Fig. 1a). Mutations in the template and daughter strands **of the fully filled plasmids** are then determined by an adapted and highly optimised version of Maximum Depth Sequencing (MDS)³⁷.

Added to the Extended Data Fig. 7 legend:

Of note, only fully filled molecules can undergo successful restriction digest and downstream library preparation for both the template and daughter strands, and therefore unfilled plasmids do not confound the results. In other words, by “template” we mean the template strand of the ROI *after* filling by the respective polymerases.

Another complicating factor is that Shcherbakova et al have also shown that the P301R enzyme is actually more faithful on a naked DNA template than the exo- PolE (Fig 1C, Xing et al Nat Comms 2019). This is inconsistent with the results shown here and not discussed.

We had discussed this already in the previous manuscript in the Supplementary Note “Discussion of mechanisms of P286R mutagenesis” (due to limitations on the length of the article):

We show that Pol ϵ P286R not only generates 27 times more mutations than *wt* Pol ϵ , but also 2.1 times more mutations than an exonuclease deficient enzyme. This demonstrates that P286R is not simply reducing the proofreading ability of the exonuclease domain. *In vivo*, the equivalent mutation to human P286R, when compared to a loss of Pol ϵ exonuclease activity, was also found to result in a much stronger mutator phenotype in fission yeast^{6,9} and shorter survival in mice^{11,12}. In contrast, the equivalent P301R mutation in *Saccharomyces cerevisiae* Pol ϵ was reported to produce a lower error rate than the exo-null mutant enzyme *in vitro*⁷, probably because the yeast P301R mutant enzyme retains more exonuclease activity compared to its human counterpart, however there might be other species-specific differences as well^{4,7,13}. While we cannot yet rule out that other factors, such as disruption of orchestrated repair, may be contributing to the hypermutation phenotype of Pol ϵ P286R, our data are compatible with the previously postulated hypothesis that P286R represents a gain-of-function mutation.

We have now made an additional change to the Discussion:

The fact that Pol ϵ P286R in a cell-free environment recreates the characteristic mutational pattern of POLED&MMRd cancer samples demonstrates that these mutations reflect the intrinsic error signature of Pol ϵ , independently of any additional factors, such as DNA damage, recruitment of other polymerases or accessory proteins (see Supplementary Note 9 for further discussion, **including potential species-specific differences**).

The authors claim that 100% of CpG sites are methylated. It follows then that all CpG sites on a single template molecule should also be methylated. This reviewer calculates that the p53 target site should then have eight NpCpG sites with methylated cytosines. It would be incredibly informative to see this fidelity resolved at the sequencing level. For example, what fraction of reads have 0, 1, 2, 3, etc (or even all 8) CpG>TpG changes in a single read?

We thank the reviewer for this suggestion. We have added the analysis of multiple mutations per single molecule as **Supplementary Note 7**. We observed that indeed, the PER-POLE-P286R and PER-POLE-EXO⁻ make multiple mistakes in the same molecule with slightly higher frequency than expected by chance (computed using permutation testing of the observed CpG>TpG mutations in the given sample). For example, in PER-POLE-P286R, the median expected fraction of molecules with multiple CpG>TpG mutations is 1.2e-5, while the observed frequency is 1.7-fold higher. In the entire dataset, we detected only two molecules with *three or more* CpG>TpG mutations, as expected given the extremely low probability of such a scenario. Altogether, our results suggest that when PER-POLE-EXO⁻ and PER-POLE-P286R make an error, there is an increased chance of another error happening in the

same molecule. However, the frequency of multiple CpG>TpG mutations in the same molecule are still very rare (only 0-50 molecules out of millions of molecules).

Figure legend: Number of molecules with multiple CpG>TpG polymerase errors detected by PER-seq in unmethylated (top) and methylated (bottom) samples. The violin plots represent simulated data, the red dots represent real measurements for individual samples. The permutation test P-value and fold-change of observed vs. median expected values are shown on top.

The reaction conditions for the Pol epsilon gap-filling reactions are not provided. The fidelity of all DNA polymerases is sensitive to salt, dNTP, Mg²⁺, pH conditions.

Thank you for your noticing that this paragraph has been accidentally missing in the Methods. We have now completed this:

The filling with human *wt* and mutant Pol ε was carried out in a 100μl of a buffer containing 20mM Tris pH 7.5, 10mM magnesium acetate, 0.1mM DTT, 150mg/ml BSA together with 100μM of each nucleotide, 40fM of polymerase and 40ng of the indicated DNA template. Reactions were incubated at 37°C for 30min. These are based on standard conditions originally set up by the Hurwitz lab⁴¹.

Filling reactions using bacterial polymerases were performed in 25μl using 100ng of gapped plasmid. For KapaU+ a 2x mmx including dNTPs (Roche, 7959052001) was used and the reaction was incubated at 72°C for 3 minutes. Klenow fragment (NEB, M0212S) filling conditions were, 1x NEB2, 1U Klenow fragment, 0.2mM dNTPs and elongation was performed at 37°C for 5min. Reactions were assembled on ice and transferred back to ice after elongation prior to magnetic bead purification.

As exciting as this technique potentially is, there is no control DNA synthesis on methylated DNA. It would be incredibly helpful to show that the CpG mutation spectrum is not seen with an unrelated pol, like Klenow-exo- for example.

We have now added this additional control of Klenow-exo- (PER-KLENOW- EXO⁻) for methylated DNA template. Both methylated and unmethylated samples show similar profiles with frequent A>T (A:dT)

and T>C (T:dG) errors, but low C>T (C:dA) (including CpG>TpG) errors. This is included as a new **Extended Data Figure 1f** and referenced from the Results.

Figure Legend: Strand-specific error signature of KLENOW-EXO⁻, when unmethylated and methylated template DNA was used for fill-in. Strand-specific error signatures of KLENOW-EXO⁻ and KAPA-U⁺. The error signature is computed as computed as error (nucleotide misincorporation) spectra with respect to the template 5' and 3' neighbouring bases (i.e., the template trinucleotide), measured by PER-seq and averaged across three replicates. For example, T:dG denotes misincorporation of guanine opposite thymine on the template strand.

Minor points

I believe in Fig 1D the authors mean to cite Jee et al 2016.

The reference to Lee et al. 2016 is correct. The reference number is mentioned in the legend, and it refers to this study: Lee, D. F., Lu, J., Chang, S., Loparo, J. J. & Xie, X. S. Mapping DNA polymerase errors by single-molecule sequencing. *Nucleic Acids Res* **44**, e118–e118 (2016).

The probability density function is not well explained.

We have now added an explanation that probability density function means that the vector sums to one.

Reviewer #3:

Remarks to the Author:

Given that C to T transitions in CpG dinucleotides are among the most common mutations found in human cancers and genetic diseases, the key results of this study are:

1. Pol ϵ is a major source of CpG>TpG errors.
2. Wild type Pol ϵ has a significantly higher error rate when replicating 5-methylcytosine (5mC)-containing CpG sites compared to replicating C in other contexts.
3. The researchers developed a new method called Polymerase Error Rate Sequencing (PER-seq) to measure the error spectrum of DNA polymerases in vitro. (This is very exciting on a personal level.)
4. The most common human cancer-associated Pol ϵ mutant (P286R) is found to produce an excess of CpG>TpG errors, mimicking the mutation pattern seen in tumors with this mutation and deficiencies in mismatch repair.
5. In human cancer samples, regions with higher MMR activity show fewer CpG>TpG mutations, suggesting that replication errors are a major contributor to CpG>TpG mutagenesis.
6. In most healthy tissues, Pol ϵ -derived CpG>TpG mutations likely outnumber those due to 5mC deamination.

I find the new technique to be novel and well presented. Many of the conclusions are quite novel and even surprising (see point #6 above). The data are well presented: the figures are clear; the logic is laid out well; and there is little to no overreach in the conclusions. The Methodologies are clearly stated and appear to be appropriate. The statistics are thorough and also appropriate. The Conclusions are robust and follow naturally from the data.

I would like to see some changes to Fig 1c: I suspect that the data exists and no further experiments will be required. If the probability of the same artefact at the same position is $\sim 10^{-9}$, then a version of Fig 1c with dilutions out to 10^{-9} or 10^{-10} would be illuminating. It's important to show where a new assay breaks down as well as where it shines.

We indeed show the full extent of data that is available. The reason why we did not include any dilutions below 10^{-6} in the experiment is because there would be too few molecules in the pool to detect them at the sequencing depth that we currently use: for most experiments here, we aim to sequence between 1 and 5 million well-covered molecules per sample. This means that at a dilution of 10^{-6} , we only observe at most a handful of molecules. At higher dilutions (10^{-7} or even 10^{-8}), we would have to also increase the sequencing depth, which is prohibitively expensive.

I would suggest an addition to the Discussion section:

(Note from reviewer: papers cited herein are for the benefit of the authors; many are already cited in this manuscript or in reviews cited by this manuscript; manuscript acceptance is not contingent on adding such citations to the next draft.)

The authors clearly know that mutation biases were used to assign Pol epsilon the role of primary leading strand replicase. However, this assignment has been called into question and the authors have a golden opportunity to explicitly weigh in. The biases for complementary mispairs in Fig 3c-f closely match those found in yeast with either engineered mutator pol2-M644G (Fig 2; PMID: 25217194) or

exo- pol2-04 (Supp. Fig. 2, PMID: 34887558). Such mutation biases, alongside ribonucleotide incorporation maps (PMIDs: 25622106, 25622295, 25624100, 26492137, 31488849, 36434012), were used to assign the leading/lagging responsibilities of the replicases across cerevisiae, pombe, and human nuclear genomes. Most of these biases were seen in Kunkel's earlier M13 LacZ assays. However, those assays had some artefacts due to, presumably, oxidative damage. Some believe Pol epsilon incapable of replicating the bulk of the leading strand (PMID: 26186286) and have used misinterpretations of said artefacts to call into question the comparability of in vitro/in vivo mutation spectra (PMID: 26145172). They claim that Pol epsilon is primarily an extrinsic proofreader of Pol delta errors and thus that mutations in POLEd & MMRd cancers are made by Pol delta (PMID: 37307920). Your data lacks oxidative damage artefacts and shows that the mutation signature in POLEd & MMRd cancer is fully explainable by Pol epsilon errors. This includes the T-dT mispairs that were key to the first leading strand assignment of Pol epsilon (PMID: 17615360) but were recently claimed NOT to be due to Pol epsilon (PMID: 37307920 again). You may be uniquely positioned to end the controversy once and for all. For the good of the field (and a potential boost to readership and impact), please add a brief discussion of the controversy and how your data puts the final nail in the coffin.

Thank you for your suggestion of using this opportunity to comment on this. We have now included the following paragraph in the discussion:

Our results also provide novel light on the long-discussed role of Pol ϵ in leading-strand DNA replication (PMID: 17615360, PMID: 37307920). Our study provides the first direct and detailed measurements of the Pol ϵ error signature (independent of potential artefacts or biases of the previously used LacZ assay). The PER-seq measured error signature of the human Pol ϵ matches the mutational signatures of POLEd & MMRd cancers, including the directionality of these mutations and their enrichment on the leading strand. Our study thus confirms the dominant role of Pol ϵ in leading-strand DNA synthesis.

Reviewer #4:

Remarks to the Author:

Mutations are essential for evolution. Understanding how these mutations arise would provide fundamental insights into the biological processes active in the living system. In this manuscript, the authors developed a novel method (PER-seq) to understand the type and frequency of DNA mismatches introduced by DNA Pol epsilon (both wild-type and mutant), one of the major human replicative DNA polymerases, at different DNA bases and the influence of neighbouring sequence context. They observed that the mismatches, which eventually lead to C>T mutations, are highly enriched at CpG sites, especially when the C is methylated. The results obtained from this PER-seq are corroborated with the somatic mutations spectrum observed in cancer patients with mutant DNA Pol epsilon and deficient MMR. In addition, they reported an interesting finding that the fidelity of wild-type Pol epsilon at non-methylated CpG is low, in general, suggesting that the C>T mutations at these sites could occur beyond the epigenomics modification found on CpG. Overall, the manuscript is well written and the findings support the conclusion. The following are a few minor comments/suggestions for improvement:

Major comments:

1) Figure 1a: What is the rationale for the choice of ROI length, ~200bp? Is this size sufficient to capture the DNA polymerase activity well (given the speed of replication)?

The length of the ROI is up to 300bp in total (including priming regions and barcodes), so that it is fully covered by 150bp paired-end sequencing. We have not observed substantial differences in replication errors on either end of the ROI, suggesting that the Pol ϵ error profile is similar just after initiation and before termination.

We chose natural sequences that exist in the human genome to avoid potential artefacts which could emerge in artificially designed sequence. The choice of the DNMT1 CpG island was to capture a CpG-rich region, to enable sufficient statistical power for the CpG-related questions. The choice of the TP53 ROI was based on two factors: first, the region contains sufficient numbers of CpG positions in all possible 5' contexts, and is generally complex enough to allow analysis of most possible trinucleotide contexts. Second, it is one of the most important and highly mutated regions in cancer – especially the CpG positions are frequent driver mutations. We have now better explained the rationale for the ROI choice in the Methods and added a Supplementary Note 8, where we show that loci with high error rates in the PER-seq experiment match known hotspot mutations in human cancer:

The TP53 ROI covers the entire exon 8 of the canonical transcript ENST00000269305. This exon comprises three of the top 5 TP53 deleterious mutation hotspots, including the most mutated amino acid (R273). To establish whether Pol ϵ errors might contribute to the generation of TP53 hotspot mutations, we identified deleterious TP53 mutations within the ROI region from TCGA data (https://portal.gdc.cancer.gov/analysis_page?app=MutationFrequencyApp, deleterious defined as high impact VEP or probably/possibly damaging PolyPhen prediction). For each position in the ROI, we selected the (predicted) deleterious variant with the highest allele frequency (if one exists).

Strikingly, the three best-known TP53 mutation hotspots in our ROI show extremely high Pol ϵ error rates in our PER-seq experiment (Supplementary Figure 8a). The R273H mutation, due to a C>T mutation in a CpG context, showed the highest PER-seq error rate, due to a mC:dA misincorporation. Notably, the PER-seq derived error frequency at loci that cause deleterious TP53 mutations significantly correlates with the frequency with which the corresponding mutation is seen amongst

TCGA patients (Spearman correlation $r = -0.6$, $p = 1e-6$, Fig. Supplementary Figure 8b). Finally, we searched for TP53 mutations (covered in this ROI) in POLEd samples (including POLEd&MMRd samples). We found four TP53 hotspots mutated in the POLEd samples (R273H, R273C, R282W, and R267W), and all of them had very high frequency in our PER-seq measurements (Supplementary Figure 8a, blank squares). In summary, our data suggest that Pol ϵ errors contribute to the generation of some of the most important cancer driver hotspots in the TP53 gene.

Supplementary Figure 8: Comparison of Pol ϵ errors in the TP53 ROI measured by PER-seq and mutation hotspots in TCGA. a, The average Pol ϵ errors measured by PER-seq on methylated template DNA of the TP53 ROI. Only deleterious variants are shown here. The type of base change is colour-coded. The top 15 deleterious TP53 hotspots (across the entire gene) are denoted by black star, and the top 120 hotspots by grey star. Variants detected in six POLEd&MMRd samples in this study are denoted by empty square. The annotations above the top hotspot bars represent: the amino acid change, sequence context, base-change, and deleterious TP53 hotspot rank. b, The PER-seq error frequency plotted against the order (rank) of deleterious TP53 hotspots in TCGA (lowest rank represents highest frequency in TCGA).

2) Figure 1d, the range of mutation frequency (from Lee et al., 2016) is not overlapping for all mutation types. For example, the range of C>T and G>T are quite high in the previous study than the PER-seq.

The assay by Lee et al. is expected to have a higher background error rate, as the assay cannot distinguish between amplification errors and true mutations, due to the absence of a linear amplification step (that is present in PER-seq). Also, their assay contains several steps of long heating at high temperature (including 98°C for 30s and other steps), which will lead to DNA damage and ultimately some false-positive mutations undistinguishable from true-positive mutations. The typical expected background errors from such damage are C>T mutations (deamination of cytosine) and G>T

(oxidation of guanine), which are exactly the mutation types higher in Lee et al. than PER-seq (Fig. 1d), and likely to explain/contribute to the difference.

3) Figure 2a-b: although it is interesting to see that the mutation profile of the PER-POLE-P286R closely resembles (with high Cosine similarity) with the mutation profiles of POLEd & MMRd tumours. However, in tumours, the mutation profile was probably shaped by deficiency of both POLE and MMR at different time points during the clonal evolution of the cancers. In other words, the deficiency of MMR occurred before POLE, or vice versa, but based on the similarity with the PER-POLE-P286R does it suggest that the tumours shown here harboured MMRd first and then POLEd?

Yes, that is correct. We have now added a Supplementary Note 6, where we address this question:

We have shown that the PER-POLE-P286R error signature closely resembles mutational profile of patients with combined POLEd&MMRd. However, it is known that the order of MMR loss and acquisition of the POLEd variant results in slightly different mutational profiles^{5,6}. Comparing our PER-seq measurements with profiles of tumours with known order of MMR loss and POLEd variant⁶, we show that PER-POLE-P286R best corresponds to profiles of cancer samples where MMR loss precedes POLEd variant (Supplementary Figure 6).

Indeed, seven of the 17 POLEd & MMRd samples included in our study have a germline biallelic MMR deficiency (bMMRd), and thus the MMR loss preceded acquisition of the POLEd variant in them. Additional six samples have a stop-gained mutation in one of the MMR genes (*MSH6*, *PMS2*, *MSH2*, or *MLH1*) with VAF higher than (or in one case similar to) the VAF of the POLE variant, in line with the MMRd preceding POLEd. Of the remaining four samples, two had high (>50%) and one fairly high (27%) *MLH1* promoter methylation, and one did not have methylation values available. In these four samples, it is hard to determine the order of the MMRd loss, however, the rest of the cohort support the conclusion that the PER-POLE-P286R error signature best resembles mutational profile of MMRd loss preceding POLEd mutation.

Supplementary Figure 6: Comparison of the PER-POLE-P286R error signature with mutational profiles of POLEd&MMRd samples by the order of MMR loss and POLEd variant acquisition. *a*, Profile of samples where MMR loss occurred first. *b*, Profile of samples where POLEd variant occurred first. *c*, Error signature of PER-POLE-P286R. Values for *a* and *b* are based on the Fig. 5A of the study by Campbell et al.⁶

4) For the above analysis, it would also be interesting to see mutation profiles in tumours with germline biallelic mismatch repair deficiency (bMMRd) followed by somatic POLE deficient events (eg., <https://doi.org/10.1038/ng.3202>).

We have now clarified in the manuscript that 7 of the 17 POLEd & MMRd samples are bMMRd from <https://doi.org/10.1038/ng.3202>. Unfortunately, the released data for this Nature Genetics study is missing the whole-genome data for all but 2 patients, and the authors did not reply to our repeated request for the data, so we could include only exome sequencing data for five of these samples.

Minor comment:

1) In Figure 1c, please specify what is being plotted on the x- and y-axis labels, is it observed (or expected) mismatches or mutations?

The artificially mutated plasmids have mutations (not mismatches). We have now clarified this in the figure legend:

The observed vs. expected frequencies of **plasmids with artificially introduced mutations**, spiked in predefined ratios (see Methods).

2) In the Methods section, under DNA polymerases: “hon-human” -> “non-human”

Thank you for spotting this typo, we have corrected this now.

Decision Letter, first revision:

9th Jul 2024

Dear Dr. Kriaucionis,

Thank you for submitting your revised manuscript "Human DNA polymerase ϵ is a source of C>T mutations at CpG dinucleotides" (NG-A62899R). It has now been seen by the original referees and their comments are below. The reviewers find that the paper has improved in revision, and therefore we'll be happy in principle to publish it in Nature Genetics, pending minor revisions to satisfy the referees' final requests and to comply with our editorial and formatting guidelines.

Sincerely,
Chiara

Chiara Anania, PhD
Associate Editor
Nature Genetics
<https://orcid.org/0000-0003-1549-4157>

Reviewer #1 (Remarks to the Author):

The authors have addressed all of my comments. I want to congratulate them with a great and important paper.

I have two small points that can still be addressed:

- Typo on page 5, line 5: "the" is missing before ability.

- Figure 4e is referred to quite late in the text and not at the point where the rest of the figure is discussed. Nevertheless, it would be good to discuss it shortly also there, as it is an interesting figure describing the PROF samples, which are missing from panel 4c.

Reviewer #2 (Remarks to the Author):

The authors have done a commendable and thorough job responding to a number of review concerns.

In particular the addition of the novel cell line analysis and mouse tumor re-analysis in the new Fig 5 is particularly helpful. It is not needed for the current manuscript, but one interesting point to be reconsidered in light of these data is the tumor discrepancy between mice with Pole-P286R and Pole-D275A/E277A. As heterozygous mice (similar to human tumors), the former readily develop tumors with the spectra re-analyzed by the current manuscript while the latter do not. The data from figure 3 provide strong evidence that it is not the nature of the mismatches. The 2-fold difference in overall mismatch frequency also fails to provide a strong rationale.

Reviewer #3 (Remarks to the Author):

My initial impression of this manuscript was quite positive and the authors have only improved it. I recommend acceptance.

Reviewer #4 (Remarks to the Author):

The authors have adequately addressed all my comments.

Author Rebuttal, first revision:

Reviewer #1 (Remarks to the Author):

The authors have addressed all of my comments. I want to congratulate them with a great and important paper.

I have two small points that can still be addressed:

- Typo on page 5, line 5: "the" is missing before ability.

We have added "the" to the indicated place of the manuscript.

- Figure 4e is referred to quite late in the text and not at the point where the rest of the figure is discussed. Nevertheless, it would be good to discuss it shortly also there, as it is an interesting figure describing the PROF samples, which are missing from panel 4c.

We added reference to Fig. 4e on page 7 of the manuscript, where the rest of the Fig. 4 is discussed.

Reviewer #2 (Remarks to the Author):

The authors have done a commendable and thorough job responding to a number of review concerns. In particular the addition of the novel cell line analysis and mouse tumor re-analysis in the new Fig 5 is particularly helpful. It is not needed for the current manuscript, but one interesting point to be

reconsidered in light of these data is the tumor discrepancy between mice with Pole-P286R and Pole-D275A/E277A. As heterozygous mice (similar to human tumors), the former readily develop tumors with the spectra re-analyzed by the current manuscript while the latter do not. The data from figure 3 provide strong evidence that it is not the nature of the mismatches. The 2-fold difference in overall mismatch frequency also fails to provide a strong rationale.

We thank the reviewer for this suggestion and interesting point. Also, we agree that this is not needed for this manuscript and will constitute part of our future work to examine different mutants of Pol ϵ .

Reviewer #3 (Remarks to the Author):

My initial impression of this manuscript was quite positive and the authors have only improved it. I recommend acceptance.

Reviewer #4 (Remarks to the Author)

The authors have adequately addressed all my comments.

Final Decision Letter:

11th Sep 2024

Dear Dr. Kriaucionis,

I am delighted to say that your manuscript "Human DNA polymerase ϵ is a source of C>T mutations at CpG dinucleotides" has been accepted for publication in an upcoming issue of Nature Genetics.

Your paper will be published online after we receive your corrections and will appear in print in the next available issue. You can find out your date of online publication by contacting the Nature Press Office (press@nature.com) after sending your e-proof corrections.

Please note that *Nature Genetics* is a Transformative Journal (TJ). Authors may publish their research with us through the traditional subscription access route or make their paper immediately open access through payment of an article-processing charge (APC). Authors will not be required to make a final decision about access to their article until it has been accepted. Find out more about Transformative Journals

Authors may need to take specific actions to achieve compliance with funder and

institutional open access mandates. If your research is supported by a funder that requires immediate open access (e.g. according to Plan S principles) then you should select the gold OA route, and we will direct you to the compliant route where possible. For authors selecting the subscription publication route, the journal's standard licensing terms will need to be accepted, including <https://www.nature.com/nature-portfolio/editorial-policies/self-archiving-and-license-to-publish>. Those licensing terms will supersede any other terms that the author or any third party may assert apply to any version of the manuscript.

If you have not already done so, we strongly recommend that you upload the step-by-step protocols used in this manuscript to [protocols.io](https://www.protocols.io). [protocols.io](https://www.protocols.io) is an open online resource that allows researchers to share their detailed experimental know-how. All uploaded protocols are made freely available and are assigned DOIs for ease of citation. Protocols can be linked to any publications in which they are used and will be linked to from your article. You can also establish a dedicated workspace to collect all your lab Protocols. By uploading your Protocols to [protocols.io](https://www.protocols.io), you are enabling researchers to more readily reproduce or adapt the methodology you use, as well as increasing the visibility of your protocols and papers. Upload your Protocols at <https://www.protocols.io>. Further information can be found at <https://www.protocols.io/help/publish-articles>.

Sincerely,
Chiara

Chiara Anania, PhD
Associate Editor
Nature Genetics
<https://orcid.org/0000-0003-1549-4157>